# FMC: Formalization of Natural Language Mathematical Competition Problems

## Abstract

Efficient and accurate autoformalization methods, which leverage large-scale datasets of extensive natural language mathematical problems to construct formal language datasets, are key to advancing formal mathematical reasoning. In this paper, we propose an autoformalization pipeline based on large language models with error feedback for syntax verification and cost-free problem decomposition for semantic alignment check, achieving a fully automatic and training-free formalization approach. Using this pipeline, we curate an Olympiad-level dataset aligning natural language problems with Lean formalizations. The dataset contains $3,214$ natural language mathematical problems and $6,994$ corresponding Lean statements, indicating a one-to-many relationship where a single problem may map to multiple formal representations. This dataset is well-suited as a benchmark for automated theorem provers. Additionally, we investigate the formalization and reasoning capabilities of various LLMs and empirically demonstrate that problem decomposition, few-shot learning and error feedback are key components to enhance the autoformalization process. Experiments of three automated theorem provers on the *FMC* dataset also highlight its challenging nature and its value as a benchmark for formal reasoning tasks.

## 1 Introduction

Large language models (LLMs), due to their strong textual reasoning capabilities, have been widely applied to mathematical problem reasoning. Initially developed for reasoning within natural language, LLMs face challenges such as the scarcity of complex mathematical data and the occurrence of hallucinations. To address these issues, formal languages have been introduced into LLM mathematical reasoning.

A formal language is a logical system in which statements and derivations can be verified through an interactive theorem prover, thereby mitigating the hallucination problem. However, formal reasoning introduces a new challenge—an even greater scarcity of data. To address this challenge, research on formal mathematical reasoning has primarily followed two directions: *automated theorem proving* and *autoformalization*, with the latter often serving as a source of training data for the former. This work focuses on the autoformalization of mathematical problems presented in natural language.

We propose an enhanced autoformalization pipeline with error feedback and problem decomposition. Building on the standard stages of *formal translation – syntax verification – semantic alignment check*, our pipeline improves accuracy using training-free general LLMs. Verification errors are collected and fed back to the formalization model, enabling self-correction. In addition, we propose a novel semantic alignment checking method, in which natural language problems are first decomposed into *data types*, *conditions*, and *proof goals*, and then compared with their formal counterparts. Compared to other decomposition method, our semantic alignment checking method needs no intermediate language which greatly decreases construction effort.

Using this pipeline, we construct a dataset of aligned natural language–Lean pairs, focusing specifically on mathematical problems of Olympiad difficulty. The original natural language problems are sourced from the website `IMOmath`, which curates problems from various national and international Olympiad competitions. After preprocessing, the problems are passed through our formal-

ization pipeline, yielding a dataset of $3,214$ natural language problems aligned with $6,994$ formal statements.

In evaluating the pipeline, we adopt *syntactic validity* and *semantic consistency* as key metrics. Our method achieves a syntactic validity of $93.39\%$ and semantic consistency of $66.99\%$. Compared to a related recent work `StepFun-Formalizer` (Wu et al., 2025), our dataset is based on significantly more challenging problems and outperforms it's autoformalization pipeline, which achieved $40.5\%$ semantic consistency.

**The contributions are summarized as follows:**

1. We propose an enhanced autoformalization pipeline using LLMs with error feedback and cost-free problem decomposition, enabling a fully automated, training-free formalization process.

2. Using this pipeline, we construct a dataset of $3,214$ aligned natural language–Lean pairs, with problems sourced from national and international mathematics Olympiads. The semantic consistency of the formalization pipeline achieves $66.99\%$, surpassing StepFun-Formalizer.

3. We investigate the formalization and semantic alignment checking capabilities of different general-purpose LLMs and find that `DeepSeek-R1` remains at the forefront. Experimental results further demonstrate that problem decomposition, few-shot learning and error feedback enhance autoformalization performance.

## 2 RELATED WORK

### 2.1 LARGE LANGUAGE MODELS

Large Language Models (LLMs) represent a significant paradigm shift in the evolution of natural language processing. Typically built upon the Transformer architecture and equipped with tens or hundreds of billions of parameters, LLMs are trained on massive textual corpora. Representative models include PaLM(Chowdhery et al., 2023), GPT-4(OpenAI, 2024), DeepSeek-V3(DeepSeek-AI, 2025b) and Claude 3.7(Anthropic, 2025). Their unprecedented scale in both model size and training data enables capabilities distinct from smaller models—capabilities often referred to as emergent abilities(Wei et al., 2022) These include in-context learning, instruction following, and step-by-step reasoning(Zhao et al., 2025), allowing LLMs to handle complex tasks across diverse domains, including reasoning tasks.

Among various reasoning tasks, mathematical reasoning has attracted particular attention. Previous research (Yang et al., 2024) has shown that using LLMs for mathematical reasoning generally involves three stages: pretraining, fine-tuning with structured data, and invoking external tools. However, reasoning within the scope of natural language using LLMs still faces significant challenges, such as the scarcity of high-difficulty data and the problem of hallucination.

As a result, some researchers have turned to formal languages for mathematical reasoning. Although data for formal languages are even more limited, their verifiability ensures the correctness and reliability of reasoning. Formal mathematical reasoning has thus become a research focus within the broader field of AI reasoning.

### 2.2 FORMAL LANGUAGES

Formal languages express mathematics within formal systems. They impose strict syntactic rules, and operations such as verification must adhere to logically sound inference rules(Yang et al., 2024).

Currently, formal math languages such as Isabelle (1986) (Paulson, 1988), Coq (1989) (Paulin-Mohring, 1993), and Lean (2015) (de Moura et al., 2015) have attracted significant attention from researchers. This study uses Lean (Moura & Ullrich, 2021), a modern open-source theorem prover developed by Microsoft Research and CMU. Lean combines interactive and automated theorem proving, and supports reasoning in both mathematics and complex systems in computer engineering.

`Mathlib` is commonly used in formalizing mathematical theorems in Lean. It is a community-driven project aimed at building a unified library of formalized mathematics for the Lean prover. `Mathlib4`, the updated version for Lean 4, includes many important mathematical objects, pre-formalized theorems, and automation strategies. Some of its metaprograms enable non-trivial proof automation.

## 2.3 AUTOMFORMALIZATION

Autoformalization, translating natural language mathematics into formal statements, is a key research direction in formal mathematical reasoning. Early studies(Wu et al., 2022) indicate that general-purpose LLMs possess a degree of formalization ability, and training with formal proof data significantly improves the performance of automated theorem provers. Survey (Weng et al., 2025) offers an in-depth investigation of autoformalization scenarios and workflows. At present, the cutting-edge theorem provers include DeepSeek-Prover-V2(Ren et al., 2025), Kimina-Prover(Wang et al., 2025), Seed-Prover(Chen et al., 2025) and Goedel-Prover-V2(Lin et al., 2025).

This work investigates how to construct a high-quality, aligned dataset of natural language and formal language pairs via autoformalization, to serve as a benchmark for evaluating theorem provers.

Constructing a Lean-based dataset via autoformalization involves two primary steps: sourcing the original data and conducting formalization. Data sources can be broadly classified into three categories: manual curation, natural language datasets, and automatic synthesis. Manual curation refers to the direct authoring of Lean theorems and proofs by mathematical experts. Natural language datasets such as the MATH dataset(Hendrycks et al., 2021), GSM8K(Cobbe et al., 2021), and AQuA-RAT(Ling et al., 2017) contain mathematical problems in natural language, which can be formalized into formal language datasets. Automatic synthesis involves the computational generation of new mathematical problems based on existing concepts and theorems. Representative works include MUSTARD(Huang et al., 2024) and STP_Lean(Dong & Ma, 2025). As for formalization methods, they can be roughly divided into two types: manual annotation, as demonstrated by works such as PutnamBench(Tsoukalas et al., 2024), miniF2F(Zheng et al., 2021), and ProofNet(Azerbayev et al., 2023); and autoformalization, which is now predominantly powered by large language models.

Our analysis of existing datasets reveals the following key observations: (1) Datasets containing competition-level problems are usually manually annotated and remain relatively small in scale. (2) Extracting natural language problems from the web and autoformalizing them is still the dominant dataset construction strategy. However, the difficulty levels of these problems frequently exhibit considerable variability. (3) Data synthesis allows for large-scale dataset generation, but data difficulty can still be improved.

Manual annotation is labor-intensive and prohibitively expensive, while typical data synthesis requires substantial computational resources. Therefore, this study aims to construct high-quality datasets automatically from natural language datasets with minimal training costs. For data selection, we focus on Olympaid-level mathematical problems to ensure sufficient difficulty and quality in the resulting dataset.

## 3 AUTOFORMALIZATION PIPELINE DESIGN

This paper proposes an autoformalization pipeline that translates mathematical problems from natural language into Lean language with error feedback and problem decomposition. The entire pipeline is shown in Figure 1. Each natural language mathematical problem is first translated into Lean language using a few-shot prompting approach and then formally verified by Lean REPL, i.e., Lean's syntax check. Those that pass the syntax verification are then compared with their decomposed original problems for semantic alignment check. Statements that pass both the syntax verification and semantic alignment check are regarded as successfully formalized. In cases where a statement fails syntax verification, the corresponding error information is incorporated into a revised prompt and fed back to the translation model, enabling iterative prompt refinement driven by error feedback. And in cases where a statement fails semantic alignment check, it will be sent back to pass through the formalization pipeline again. Prompts are listed in Appendix D.

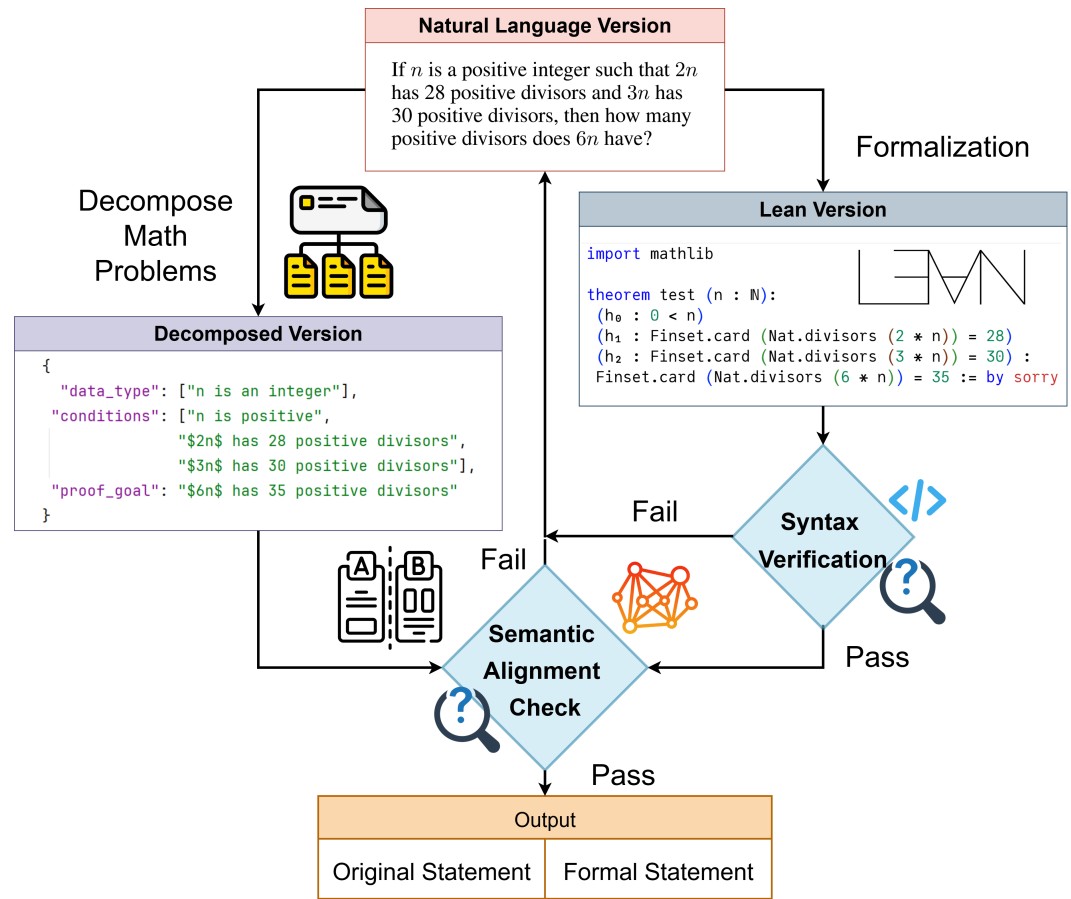

Figure 1: The FMC autoformalization pipeline consists of three stages: (1) formalizing the original problem from natural language into Lean; (2) verifying Lean syntax using the Lean REPL and feeding back syntax errors for retranslation; and (3) performing semantic alignment checking by decomposing the original problem into data_type, conditions, and proof_goal, and comparing this structure with the formalized statements. Problems failing either syntax verification or semantic alignment are retranslated.

### 3.1 TRANSLATION

Within this autoformalization pipeline, the model is required to translate natural language mathematical problems into formal language in two distinct scenarios: the first involves direct translation guided by few-shot prompting, while the second leverages compiling error feedback incorporated into the prompt following a failed attempt.

The first case occurs during the initial translation of a mathematical problem and is based on few-shot learning. In this study, each translation prompt includes two fixed examples—one from algebra and one from number theory—both correctly aligned in natural language and Lean. Since problems in these two fields are common in math Olympiad contests, using such examples helps improve translation accuracy. This pipeline sets the temperature to 1.0 during formalization and samples each input five times.

The second case arises when a theorem fails syntax verification. In such instances, the associated error information is incorporated into the prompt and fed back to the translation model. The model then attempts to retranslate the theorem, leveraging the in-context learning capabilities of LLMs to produce a valid formal representation. Experimental results demonstrate that incorporating compiling error feedback improves formalization accuracy, as detailed in the experimental section.

Surprisingly, we empirically find that semantic errors is not conducive to formalization accuracy, as demonstrated in the experiment described later. So the semantic misaligned statements is passed

through the whole pipeline again without error feedback. This approach is roughly equivalent to increasing the sampling number, but it reduces cost by re-translating only semantic misaligned statements.

Additionally, since Lean focuses on theorem proving and cannot resolve open problems lacking explicit solutions within the statements, it is necessary to address missing solutions and proofs. For absent solutions, we depend on the model's reasoning capabilities, expecting it to generate them during the formalization process. For missing proofs, the placeholder ":= by sorry" is employed, enabling Lean to detect the omission and signal the missing proof without triggering errors.

After comparing formalization capabilities of several frontier language models, Deepseek-R1 ranks first. Experiments are presented in the Section 5.1. Accordingly, we adopt Deepseek-R1 (DeepSeek-AI, 2025a) as our translation model. As one of the state-of-the-art large language models, Deepseek-R1 possesses strong reasoning and in-context learning abilities, demonstrating promising performance in formalization.

### 3.2 SYNTAX VERIFICATION

Each translated formal theorem must undergo syntax verification to ensure its syntactic correctness. In this study, the formal verifier from DeepSeek-Prover(Xin et al., 2024) is used to submit the Lean statement to Lean 4 REPL for validation and parsing the returned results. Specifically, syntax verification is performed by invoking Lean 4's interactive Read-Eval-Print Loop (REPL) via a Python subprocess. The formal theorem, including Mathlib import statements, is passed through standard input in JSON format, while the output records any compilation or type errors. This approach confirms the syntactic correctness of the formal theorem and provides error messages for failed compilations, facilitating iterative prompt refinement based on error feedback.

### 3.3 SEMANTIC ALIGNMENT

Even when statements formalized by LLMs pass Lean's syntax checks, they sometimes differ semantically from the original theorems—such as missing conditions, incorrect assumptions, or erroneous goals. These discrepancies clearly indicate incorrect formalization. Therefore, semantic verification is necessary to ensure that the formalized theorems correspond to the same mathematical problems described in the original statements.

Inspired by work KELPS(Zhang et al., 2025), we have the LLM simulate the human thinking process in semantic alignment checks. It first decomposes the natural language problem into three parts: *data types*, *conditions* and *proof goals* to prevent confusion between conditions and conclusions, and then compares whether the semantics of the decomposed problem are consistent with the formal statement. But different from KELPS, our decomposition method introduces no intermediate language and thus is simpler and more lightweight.

The reason why natural language problems instead of formal statements are decomposed is that the expression of natural language mathematical problems is usually more ambiguous and complex so that there may be implicit conditions and unclear proof objectives. In contrast, the conditions and goals in Lean are more distinct. Decomposition splits semantic alignment into two distinct tasks – understanding and comparison, and enables LLMs to focus on one task at one time. Empirical results show that decomposing natural language problems enhances formalization quality, surpassing both direct semantic alignment checks and comparisons with back-translated problems. Experiments are shown in Section 5.2.

Notably, during the process of decomposing math problems, we also adopt a few-shot approach to help the model better understand the distinctions among data type, conditions, and proof goal, while ensuring a unified output format. In addition, the prompt explicitly instructs the model to compute an answer for problems without one and use it as the proof goal, thereby completing the formulation of the theorem.

Similarly, this automated pipeline employs Deepseek-R1(DeepSeek-AI, 2025a) as the consistency checking model. Notably, due to Deepseek-R1's strong reasoning capabilities, it often produces extensive analytical output. To facilitate the extraction of relevant information, the prompt explicitly restricts the output format.

Table 1: Result of formalization

| CLASS | NUMBER | RATIO |
|---|---|---|
| Total | 4798 | 100% |
| Syntax verification | 4481 | 93.39% |
|     Pass at one go | 4287 | 89.35% |
|     Pass with error feedback | 194 | 4.04% |
| Consistency check | 3214 | 66.99% |
|     Pass at one go | 2426 | 50.56% |
|     Pass with retranslation | 788 | 16.42% |

## 4 DATASET CONSTRUCTION

### 4.1 DATA COLLECTION

To enhance dataset difficulty and ensure quality, all mathematical problems in this dataset are all official Olympiad problems on the `IMOmath` website(IMOmath, 2025). It covers 11 international competitions including the International Mathematical Olympiad (IMO), as well as 42 national and regional Olympiad contests. The dataset spans the years 1959 to 2011 and covers six continents. By selecting these sources as the original natural language data, the dataset guarantees the integrity, correctness, and challenge of the mathematical theorems. The resulting aligned natural language–Lean dataset will facilitate the evaluation of automated theorem provers' capabilities.

To collect problems, this study employed web crawling techniques to download PDF files from relevant websites, and used Optical Character Recognition (OCR) to extract problem statements from the PDFs. In this work, we used Mathpix (Huang et al., 2023) as the OCR tool to convert PDF content into markdown files. Finally, we used regular expressions to extract the problem texts from the markdown files and organized them into JSON format. After preprocessing, we obtained a total of 6,980 natural language mathematical problems.

### 4.2 DATA PREPROCESSING

To further improve the quality of the formalized data, this study conducted additional filtering on the extracted $6,980$ natural language mathematical problems. Preliminary experiments revealed that sometimes geometric problems can pass both the syntax verification and consistency checks in the autoformalization pipeline, but are still wrong formalization examples. This indicates that LLMs have limited understanding of geometric problems, as shown in Appendix E. Although some prior work has attempted autoformalization of Euclidean geometry problems(Murphy et al., 2024), even with improvements, the accuracy remains around $20\%$. Therefore, this study temporarily excludes geometry problems for better dataset quality and reduce the burden for prohibitively expensive expert annotation.

The dataset also contains some mathematical problems where a single problem includes multiple subproblems.Although the Lean system can handle multi-goal problems, for simplicity, multiple subproblems under the same problem number are split. Specifically, the original problem is divided into subproblems of the form "shared conditions + subgoal 1," "shared conditions + subgoal 2," and so forth.

The tasks of filtering out geometry problems and splitting subgoals are also performed by the Deepseek-R1 model. After these processes, a total of $4,798$ natural language mathematical problems were retained.

### 4.3 DATASET CONSTRUCTION AND EVALUATION

The $4,798$ natural language mathematical problems were processed through the autoformalization pipeline in this study. After multiple sampling rounds and error feedback iterations, $4,481$ formalized statements passed syntax verification, achieving a pass rate of $93.39\%$. Subsequently, after semantic alignment check, a natural language–Lean aligned dataset comprising $3,214$ entries was constructed, corresponding to a formalization accuracy of $66.99\%$.

Table 2: Model performance comparison. The number of tokens and the time are both average values.

| MODEL | SYNTAX VERIFICATION | | | ALIGNMENT CHECK | | |
|---|---|---|---|---|---|---|
| | TOKEN NUMBER | TIME(S) | PASS RATE | TOKEN NUMBER | TIME(S) | PASS RATE |
| Deepseek-R1 | 10 783.79 | 338.16 | 58% | 3357.41 | 94.74 | 43% |
| GPT-4o-mini | 1508.33 | 17.00 | 34% | 737.20 | 3.97 | 11% |
| Claude 3.7 Sonnet | 1721.87 | 43.28 | 31% | 881.16 | 25.10 | 27% |

Among the 4,481 statements that passed syntax verification, 4,287 were verified on the first attempt, while 194 were corrected and passed after error feedback, improving the syntax verification pass rate by $4.04\%$. Of the $3,214$ entries that passed semantic alignment check, $2,426$ passed initially, and 788 were corrected and passed after second round translation, increasing the consistency check pass rate by $16.42\%$. Detailed data are presented in Table 1. These results demonstrate that automated error feedback is highly effective in improving formalization accuracy.

To better evaluate the *FMC* dataset, it is compared with other datasets in terms of domain distribution and formalization styles, with detailed results presented in Appendix B. Furthermore, to assess the quality of the *FMC* dataset, an additional 50 samples are randomly selected as a subset and are conducted a new manual verification on, yielding an accuracy of 78%. The detailed error distribution and case studies are provided in Appendix C.

## 5 EXPERIMENTS

### 5.1 AUTOFORMALIZATION CAPABILITY OF DIFFERENT LLMS

This paper compares the formalization capabilities of Deepseek-R1, GPT-4o-mini, and Claude 3.7 Sonnet. The selection of these three models primarily considers their strengths in reasoning and their high efficiency. They are respectively applied as the formalization model and semantic alignment check model within the autoformalization pipeline. Experiments were conducted on a random sample of 100 original problems (including geometry problems) and follows the pipeline in Lean Workbook(Ying et al., 2024). The results, shown in Table 2, indicate that at the cost of higher resource use and time, Deepseek-R1 significantly outperforms the other two models in formalization capability, achieving a final formalization accuracy of 43%. Although the theorems formalized by GPT-4o-mini and Claude 3.7 Sonnet exhibit comparable pass rates in syntax verification, the final number of theorems passing consistency check is substantially higher for Claude 3.7 Sonnet than for GPT-4o-mini.

Table 3: Pass rates of different formalization models.

| | DEEPSEEK-R1 | GPT-4O-MINI | CLAUDE 3.7 SONNET |
|---|---|---|---|
| Formalization pass rate | 58% | 34% | 31% |
| Consistency check pass rate | 43% | 10% | 22% |

To further investigate the formalization capabilities of the three models, experiments were conducted with the same semantic alignment check model (Deepseek-R1). The experimental results are presented in Table 3.

It is observed that when using the same alignment checking model, Deepseek-R1 achieves significantly higher syntax verification and semantic alignment check pass rates. This indicates that Deepseek-R1 is better at ensuring both syntactic and semantic correctness during formalization. Although GPT-4o-mini's syntax verification pass rate is slightly higher than that of Claude 3.7 Sonnet, its semantic alignment check pass rate is notably lower. This suggests that while GPT-4o-mini adheres to Lean's syntax rules, it struggles to accurately capture the original mathematical problem's intent.

To further investigate the formalization and reasoning capabilities of Deepseek-R1 and identify its limitations, statistics and case studies are conducted on 50 randomly selected statements that fail the semantic alignment check. The results are presented in Appendix A.

## 5.2 Effect of different semantic alignment checking methods and LLMs

To efficiently verify the semantic consistency between formal statements and their corresponding natural language math problems at scale, we employ LLMs for semantic checking. Two approaches are explored. The first follows the Lean Workbook(Ying et al., 2024) Natural Language Inference (NLI) methodology: formal statements are back-translated into natural language, after which LLMs are used to check consistency between the back-translation and the original problem. The second approach, inspired by KELPS(Zhang et al., 2025), decomposes the original natural language problem into three components—data type, conditions, and proof goal—and then asks the model to judge whether the decomposition remains faithful to the original problem. We also evaluate the capabilities of three specific models: Deepseek-R1, GPT-4o-mini, and Claude 3.7 Sonnet. We manually inspected a set of randomly selected samples and annotated whether their formal statements were semantically aligned with the original problems. We then compared the judgments made by the large language model with the expert annotations, considering a judgment correct if it agreed with the annotation and incorrect otherwise. Accuracy, precision, recall, and F1 score are adopted as evaluation metrics, and experiments are conducted on a randomly selected subset of problems. The results are shown in Table 4.

Table 4: Evaluation metrics for semantic consistency checks using two methods: natural language inference (NLI) and problem decomposition. All evaluation data are derived from the formalizations generated by Deepseek-R1.

| Model Name | Accuracy | Precision | Recall | F1 score |
|---|---|---|---|---|
| Deepseek-R1 (NLI) | 69.0% | 66.0% | 95.0% | 78.0% |
| GPT-4o-mini (NLI) | 69.0% | 71.0% | 77.0% | 74.0% |
| Claude-3-7 (NLI) | 62.0% | 59.0% | 100.0% | 75.0% |
| Deepseek-R1 (Decomp.) | 82.0% | 80.0% | 91.0% | 85.0% |
| GPT-4o-mini (Decomp.) | 64.0% | 72.0% | 59.0% | 65.0% |
| Claude-3-7 (Decomp.) | 72.0% | 70.0% | 86.0% | 78.0% |

The results show that, under the "back-translation - semantic alignment checking" method, Claude 3.7 Sonnet tends to give affirmative judgments, frequently classifying mathematically inconsistent natural language–Lean pairs as correct. This leads to very high recall. In contrast, GPT-4o-mini applies stricter criteria for consistency, achieving higher precision by producing more reliable positive predictions. Deepseek-R1 strikes the best balance between precision and recall, yielding the strongest overall performance. Under the "problem decomposition - semantic alignment checking" method, Deepseek-R1 clearly outperforms the other two models across all four metrics. This suggests that for decomposition tasks—which demand stronger reasoning ability—Deepseek-R1 holds a significant advantage. Overall, the "problem decomposition – semantic alignment checking" approach with Deepseek-R1 proves most effective, aligning closely with human judgment. Consequently, this method is adopted in our work for semantic consistency verification.

## 5.3 Effect of Few-shot Learning

In this experiment, few-shot learning was applied both to formalization and semantic consistency checking. Specifically, two example translations were provided when converting mathematical problems from natural language into Lean, and decomposition examples were also included in subsequent problem decomposition tasks. Experimental results in Table 5 indicate that few-shot learning improves both the syntax accuracy of formalizations and their semantic consistency. For instance, few-shot learning increased the syntax verification pass rate from 79.0% to 89.5%, and the semantic alignment pass rate from 29.3% to 30.0%. This demonstrates that few-shot learning is more effective at enhancing the syntactic correctness of formalizations, likely because the provided Lean code examples help the model generate compilable code. In contrast, improving semantic consistency

relies more on the model's reasoning capabilities, which explains the relatively smaller gains in this aspect.

Table 5: The effect of few-shot learning on formalization accuracy.

| METRIC | WITHOUT FEW-SHOT | WITH FEW-SHOT |
|---|---|---|
| Syntax verification pass rate | 79.0% | 89.5% |
| Semantic alignment pass rate | 29.3% | 30.0% |

## 5.4 EFFECT OF ERROR FEEDBACK

The error feedback experiments were conducted separately on syntax errors and semantic errors. Syntax errors were derived from failed syntax verification results. By including the syntax error together with the original problem in the prompt for re-translation, the syntax verification pass rate increased to 66.7%, compared to 60.0% when re-translating the original problem alone. Considering that the initial syntax verification pass rate was already high (approximately 90%), the failed cases are likely to correspond to ill-defined problems, making this improvement satisfactory.

In contrast, the effect of error feedback on semantic errors was less promising. Semantic errors were obtained from failed semantic alignment checks. Similarly, including the semantic error with the original problem in the prompt for re-translation resulted in a pass rate of 26.7%, which is lower than both re-translating the original problem alone (29.3%) and using few-shot examples (30%). This suggests that incorporating information from semantic errors may introduce noise into the formalization process. Statements with semantic errors may benefit more from diversity in formalizations rather than modifications based solely on the initial incorrect version. Experiment data is shown in Table 6

Table 6: The effect of error feedback on formalization accuracy.

| ERROR TYPE | NO ERROR FEEDBACK | WITH ERROR FEEDBACK | FEW-SHOT EXAMPLES |
|---|---|---|---|
| Syntax error | 60.0% | 66.7% | - |
| Semantic error | 29.3% | 26.7% | 30.0% |

## 5.5 TESTING AS A BENCHMARK FOR AUTOMATED THEOREM PROVERS

To assess the relative difficulty of formal mathematical datasets, we benchmark five state-of-the-art provers—Kimina-Prover, Goedel-Prover-SFT, DeepSeek-Prover-V1.5-RL, DeepSeek-Prover-V2-671B and Goedel-Prover-v2—on our newly constructed *FMC* dataset. Each prover was run 32 times on $1,000$ randomly sampled problems from the $6,994$ formal statements, with consistent hyperparameters to ensure statistical reliability. As shown in Table 7, for Kimina-Prover, Goedel-Prover-SFT and DeepSeek-Prover-V1.5-RL, the quality of *FMC* dataset is comparable to ProofNet and FormalMath. But for stronger model such as DeepSeek-Prover-V2-671B or Goedel-Prover-v2, more statements in *FMC* dataset can be proved. Nevertheless, it can still be observed that *FMC* remains substantially more challenging compared to miniF2F and FormalMath-Lite. These results confirm that our dataset provides a substantial challenge and clearly differentiates prover performance on moderately complex formal reasoning tasks, underscoring its value as a benchmark.

Table 7: Test results of different automated theorem provers. Each verification task was evaluated over 32 runs on $1,000$ randomly sampled formal problems.

| DATASET | KIMINA-PROVER | GOEDEL-PROVER-SFT | DEEPSEEK-PROVER -V1.5-RL | DEEPSEEK-PROVER -V2-671B | GOEDEL-PROVER-V2 |
|---|---|---|---|---|---|
| MiniF2F | 63.1% | 57.6% | 50.0% | 82.4% | 88.1% |
| ProofNet | - | 15.2% | 16.0% | 23.8% | - |
| FormalMATH | 16.5% | 13.5% | 10.2% | 28.31% | - |
| FormalMATH-Lite | 48.9% | 46.7% | 47.98% | 56.0% | - |
| *FMC* | 17.6% | 18.3% | 8.6% | 40.5 % | 62.5% |

# 6 CONCLUSION

This paper reviews existing natural language–Lean aligned mathematical problem datasets, noting that competition-level datasets are typically small, while automatically constructed ones often lack guaranteed difficulty. To address this gap, we introduce an autoformalization pipeline with error feedback and problem decomposition, which iteratively optimizes prompts based on generated feedback. The pipeline relies entirely on off-the-shelf general-purpose large language models, showcasing their reasoning and formalization capabilities while reducing deployment costs.

Using this pipeline, we built a natural language–Lean dataset containing $3,214$ natural language problems and $6,994$ formalized Lean statements at Olympiad-level difficulty, achieving a semantic accuracy of $66.99\%$. Ablation studies further demonstrate the advantages of Deepseek-R1 over GPT-4o-mini and Claude 3.7 Sonnet, as well as the positive impact of problem decomposition, few-shot learning, and error feedback. Finally, evaluations with three state-of-the-art theorem provers highlight the dataset's challenging nature and value as a benchmark for formal reasoning.

## USE OF LARGE LANGUAGE MODELS

In addition to the experiments mentioned in the article, large language models were also used in aiding and polishing writing of this paper.

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

## A  APPENDIX: SEMANTIC FAILURE OF AUTOFORMALIZATION

### A.1  STATISTICAL ANALYSIS

In the process of translating mathematical problems into formal statements acceptable to theorem provers, a statistical analysis of semantic failures is are made for understanding the model's limitations. 50 problems that failed the semantic consistency check are randomly selected and categorized. The results are summarized in the table 8.

Table 8: Statistics of semantic-alignment failure cases

| Problem Type | Count | Secondary Problem Type | Count |
|---|---|---|---|
| Data type error | 3 | Incorrect data type | 3 |
| Condition error | 14 | Condition too strong or too weak | 2 |
| | | Missing constraint | 6 |
| | | Incorrect constraint | 6 |
| Proof goal error | 26 | No solution provided | 13 |
| | | Incomplete solution | 4 |
| | | Incorrect solution | 3 |
| | | Incorrect proof goal | 2 |
| | | Incomplete proof goal | 1 |
| | | Direct output of few-shot example | 3 |
| Modeling failure | 4 | – | 4 |
| * Judgement error | 7 | – | 7 |

Note that judgement error refers to cases where a correctly semantic aligned statement is mistakenly classified as a semantic failure. And the total count exceeds 50 because some problems contain multiple types of errors simultaneously.

As shown in the table, semantic issues occur most frequently in proof goal errors, followed by condition errors. Among the proof goal cases, many instances without a solution were either left unsolved or solved incorrectly, highlighting one of the current challenges in automatic formalization.

Table 9: Category Distribution

| Category | Count | Percentage |
|---|---|---|
| Algebra | 21 | 21.0% |
| Analysis | 24 | 24.0% |
| Number Theory | 26 | 26.0% |
| Combinatorics | 20 | 20.0% |
| Probability | 1 | 1.0% |
| Set Theory | 1 | 1.0% |
| Other | 7 | 7.0% |

Distribution across mathematical topics are examined further. Problems without a solution are often related to finding functions or determining value ranges, while modeling failures typically occur in applied problems and combinatorics. Additionally, questions involving maxima, minima, or attainable function values are frequently overlooked during formalization. The distribution of semantic failures across mathematical domains is summarized in table 9.

## A.2 CASE STUDY

### A.2.1 MODELING FAILURE

Modeling failure occurs when formalized statements fail to accurately capture the mathematical essence of original problems. These issues are particularly prominent in combinatorial mathematics and recursive structure problems.

> **Original problem**: $n$ children sit in a circle. A teacher distributes candies clockwise starting from one child: gives to first, skips one, gives to next, skips two, gives to next, and so on (increasing skips). Find $n$ such that eventually every child receives at least one candy.
>
> **Formalized output**:
>
> ```
> theorem candy_distribution (n : ℕ) :
>   ∃ m : ℕ, ∀ i : Fin n, ∃ k : ℕ,
>   k < m ∧ (k * (k + 3) / 2) % n = i := by sorry
> ```

**Analysis**: The formalization exhibits three fundamental discrepancies. Primarily, there exists a critical mismatch in the core mathematical expression: the adopted quadratic form $(k(k+3)/2)$ fails to correspond with the actual skipping pattern, where triangular numbers $k(k+1)/2$ would be the correct characterization of the candy distribution rule. More critically, the conclusion remains incomplete as the formalization merely proves the existence of qualifying $n$ without characterizing its essential property (specifically that $n$ should be a power of two, $n = 2^k$), thereby depriving the result of practical applicability.

> **Original problem**: Given triangle with sides $a, b, c$, construct new triangle with sides $s - a, s - b, s - c$ where $s$ is semiperimeter. Repeat until construction impossible. Find initial triangles allowing infinite iteration.
>
> **Formalized output**:
>
> ```
> theorem indefinite_semiperimeter_triangle (a b c : ℝ
>   ) :
>   (∀ s : ℝ, s = (a + b + c) / 2 →
>     ∃ a' b' c' : ℝ,
>       a' = s - a ∧ b' = s - b ∧ c' = s - c ∧
>       a' + b' > c' ∧ b' + c' > a' ∧ c' + a' > b' ∧
>       ∀ n : ℕ,
>         ∃ a'' b'' c'' : ℝ,
>           a'' = (a' + b' + c') / 2 - a' ∧
>           b'' = (a' + b' + c') / 2 - b' ∧
>           c'' = (a' + b' + c') / 2 - c' ∧
>           a'' + b'' > c'' ∧
>           b'' + c'' > a'' ∧
>           c'' + a'' > b'') →
>   a = b ∧ b = c := by sorry
> ```

**Analysis**: The formalization contains three critical flaws in recursive construction. Primarily, it demonstrates insufficient recursion depth by explicitly building only two iteration levels (from $(a, b, c)$ to $(a', b', c')$ to $(a'', b'', c'')$), failing to establish a genuine infinite recursion structure. Secondly, there exists structural discontinuity between iteration levels - the inputs of the second iteration $(a', b', c')$ lack proper mathematical connection to the outputs of the first iteration $(a, b, c)$, creating an open-loop system that breaks recursive coherence.

### A.2.2 PROOF GOAL ERROR

Solving-oriented problems often suffer from missing or misplaced goals, especially when requiring specific values or function expressions.

**Original problem**: Find all functions $f : \mathbb{R} \to \mathbb{R}$ satisfying: (i) $f(x + y) \in [f(x) + f(y), f(x) + f(y) + 1]$ (ii) $x \in [0, 1) \Rightarrow f(0) \geq f(x)$ (iii) $f(1) = 1 \wedge -f(-1) = 1$

**Formalized output**:

```
theorem translated_theorem :
  ∃ f : ℝ → ℝ,
  (∀ x y : ℝ, f x + f y + 1 ≥ f (x + y) ∧ f (x + y)
   ≥ f x + f y) ∧
  (∀ x : ℝ, 0 ≤ x ∧ x < 1 → f 0 ≥ f x)
  ∧ f 1 = 1 ∧ -f (-1) = 1 := by sorry
```

**Analysis**: The primary issue is goal misalignment: while the original problem demands "finding all satisfying functions", the formalized version merely proves the existence of such functions and omits the concrete solution set, representing a substantial deviation from the original requirement.

**Original problem**: Let $\{f(n)\}$ be strictly increasing sequence of positive integers where the $n$-th missing positive integer is $f(f(n)) + 1$. Find $f(240)$.

**Formalized output**:

```
theorem test
  (f : ℕ → ℕ)
  (h₀ : ∀ n, 0 < f n)
  (h₁ : ∀ n m, n < m → f n < f m)
  (h₂ : ∀ n, ∃ m, m = f (f n) + 1 ∧ ∀ k < m, (∃ i, f
   i = k) → ∃ i ≤ k, f i = k) :
  f 240 = 38808 := by sorry
```

**Analysis**: The formalization exhibits two fundamental flaws. Firstly, it fails to properly characterize the "n-th missing positive integer" - while it expresses the relationship $m = f(f(n)) + 1$ using existential quantification, it lacks the necessary cardinality constraints to ensure this indeed represents the n-th (rather than any arbitrary) missing number. More critically, the formalized conclusion's assertion that $f(240) = 38808$ is incorrect, revealing a serious validation failure.

### A.2.3 CONDITION ERROR

Omission of critical conditions during formalization can lead to incorrect conclusions, particularly dangerous in number theory problems.

**Original problem**: Let $k \geq 14$, $p_k$ the largest prime strictly less than $k$ (given $p_k \geq 3k/4$). Let $n$ be composite. Prove: (a) If $n = 2p_k$, then $n \nmid (n - k)!$ (b) If $n > 2p_k$, then $n \mid (n - k)!$

**Formalized output**:

```
theorem part_a (k : ℕ) (hk : k ≥ 14)
  (p_k : ℕ) (hp_k : Nat.Prime p_k)
  (h : p_k < k) (h₁ : p_k ≥ 3*k/4)
```

```
        (n : ℕ) (hn : ¬Nat.Prime n)
        (hn_eq : n = 2 * p_k) :
        ¬n | (n - k).factorial := by sorry
```

**Analysis**: The formalization exhibits several critical limitations. Most notably, it fails to explicitly specify that $p_k$ must be the largest prime strictly less than $k$. While the theorem states that $p_k$ is prime and satisfies $p_k < k$, the absence of the maximality condition introduces a fundamental flaw—the conclusion may become invalid if multiple primes satisfy $p_k < k$.

## B  APPENDIX: COMPARISON WITH OTHER DATASETS

### B.1  COMPARISON OF DOMAIN DISTRIBUTION

To gain a more comprehensive understanding of the differences in mathematical domain distribution across various formalization datasets, we compare the distributions of *FMC* FormalMath, and Lean Workbook across major mathematical domains.

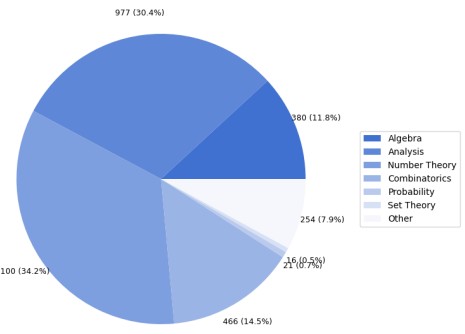

Figure 2: Distribution of mathematical domains in the FMC dataset.

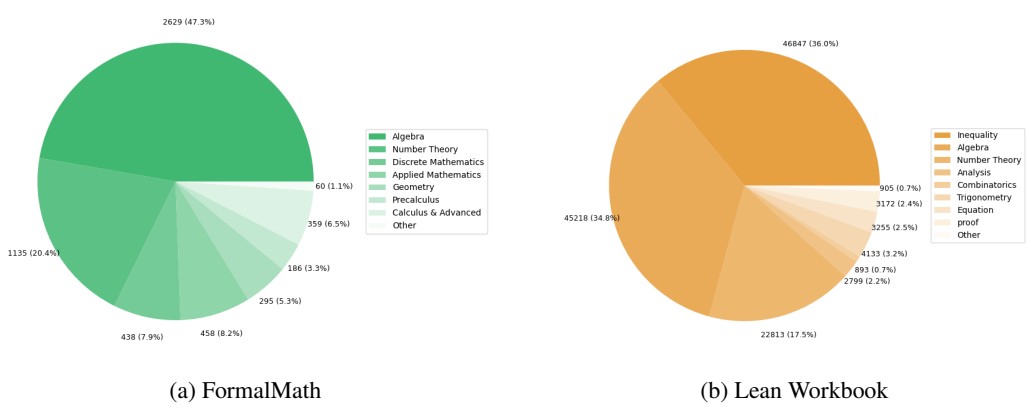

(a) FormalMath                                    (b) Lean Workbook

Figure 3: Distribution of mathematical domains in FormalMath and Lean Workbook datasets.

Figure 2 (*FMC*), Figure 3a (FormalMath) and Figure 3b (Lean Workbook) show their distribution difference. Please note that each dataset follows its own publicly defined categorization scheme, and these schemes differ across datasets. For clearer comparison, we merged several categories where appropriate; however, some datasets inherently exhibit a richer variety of labels. It is to emphasize that the diversity in category distributions arises not only from the dataset content itself, but also from

differences in their underlying classification standards. In summary, the FMC dataset demonstrates a relatively balanced and diverse distribution across mathematical domains.

## B.2 COMPARISON OF FORMALIZATION STYLES

To validate the reliability of our dataset, we selected some mathematical problems which overlap with FMC from FIMOLiu et al. (2023), CombiBenchLiu et al. (2025), and FormalMATHYu et al. (2025), and compared their formalization accuracy and stylistic differences. All three datasets adopt Lean as their formal language.

### B.2.1 FIMO

The FIMOLiu et al. (2023) dataset contains problems drawn from the International Mathematical Olympiad (IMO) Shortlisted Problems. Below is an example that appears in both FMC and FIMO.

---

**Original problem**: Prove that

$$\frac{x^2}{(x-1)^2} + \frac{y^2}{(y-1)^2} + \frac{z^2}{(z-1)^2} \geq 1$$

for all real numbers $x, y, z$, each different from 1, and satisfying $xyz = 1$.

**FMC**:

```
theorem test
  (x y z : ℝ)
  (h₀ : x ≠ 1)
  (h₁ : y ≠ 1)
  (h₂ : z ≠ 1)
  (h₃ : x * y * z = 1) :
  x^2 / (x - 1)^2 + y^2 / (y - 1)^2 + z^2 / (z -
    1)^2 ≥ 1 := by sorry
```

**FIMO**:

```
theorem fimo_2008_algebra_p2_1
  (x y z : ℝ)
  (h₀ : x ≠ 1 ∧ y ≠ 1 ∧ z ≠ 1)
  (h₁ : x * y * z = 1) :
  x^2 / (x - 1)^2 + y^2 / (y - 1)^2 + z^2 / (z -
    1)^2 ≥ 1 :=
begin
  sorry
end
```

---

This problem involves simple conditions and a clear goal so that both formal statements accurately capture the intended meaning with only minor differences. Specifically, FMC lists the assumptions $x \neq 1, y \neq 1$ and $z \neq 1$ separately, while FIMO combines them into a single conjunctive premise. From a formal reasoning perspective, FMC's formulation facilitates usage of assumptions, whereas FIMO's version offers improved readability. Additionally, the two statements' placeholder styles differ: `by sorry` versus `begin...sorry...end`, reflecting style discrepancy between Lean versions.

### B.2.2 FORMALMATH

The FormalMATHYu et al. (2025) dataset spans a wide range of topics, from high school Olympiad problems to undergraduate-level theorems. Below are two examples found in both FMC and FormalMATH.

- Case 1

> **Original problem**: Find all functions $f : \mathbb{R} \to \mathbb{R}$ such that for all real numbers $x, y$,
> $$f(f(x) + y) = f(x^2 - y) + 4yf(x).$$
>
> **FMC**:
>
> ```
> theorem test
>   (f : ℝ → ℝ)
>   (h₀ : ∀ x y, f (f x + y) = f (x ^ 2 - y) + 4 * y *
>   f x) :
>   f = 0 ∨ f = fun x => x ^ 2 := by sorry
> ```
>
> **FormalMATH**:
>
> ```
> theorem olymid_ref_base_11031 (f : ℝ → ℝ) :
>   (∀ x y, f (f x + y) = f (x ^ 2 - y) + 4 * y * f x)
>   ↔
>   ∀ x, f x = 0 ∨ f x = x ^ 2 := by
> ```

Similar to the FIMO example, both formalizations are largely consistent and accurate with respect to the natural language description.

- Case 2

> **Original problem**: Show that for any integer $n \geq 2$ the sum of the fractions $\frac{1}{ab}$, where $a$ and $b$ are relatively prime positive integers such that $a < b \leq n$ and $a + b > n$, equals $\frac{1}{2}$. (Integers $a$ and $b$ are called relatively prime if the greatest common divisor of $a$ and $b$ is 1.)
>
> **FMC**:
>
> ```
> theorem test
>   (n : ℕ)
>   (h₀ : 2 ≤ n) :
>   Finset.sum (Finset.filter (\lambda ab : ℕ × ℕ =>
>   ab.1 < ab.2 ∧ ab.2 ≤ n ∧ ab.1 + ab.2 > n ∧
>   Nat.gcd ab.1 ab.2 = 1) (Finset.product
>   (Finset.Icc 1 n) (Finset.Icc 1 n))) (\lambda ab =
>   > 1 / (ab.1 * ab.2 : ℚ)) = 1 / 2 := by sorry
> ```
>
> **FormalMATH**:
>
> ```
> theorem olymid_ref_base_11032 {n : ℕ} (hn : 2 ≤ n) :
>   Σ' a : ℕ, Σ' b : ℕ,
>   (if (a < b ∧ b ≤ n ∧ Nat.Coprime a b ∧ a + b > n)
>   then (1 / ((a * b) : ℚ)) else 0) =
>   1 / 2 := by
> ```

From a mathematical perspective, both versions express the same content rigorously. The key difference lies in stylistic preference: FMC uses `Finset` to denote finite sets in Lean, aligning with Lean's programming idioms, while FormalMATH opts for summation notation and `if...then` logic, which more closely resembles traditional mathematical expressions and improves human readability.

### B.2.3 COMBIBENCH

The CombiBenchLiu et al. (2025) dataset contains 100 carefully selected combinatorial problems formalized by mathematical experts, including all combinatorial IMO problems since 2000. Below is an overlapping example found in both FMC and CombiBench.

---

**Original problem**: A magician has one hundred cards numbered 1 to 100 . He puts them into three boxes, a red one, a white one, and a blue one, so that each box contains at least one card. A member of the audience draws two cards from two different boxes and announces the sum of numbers on those cards. Given this information, the magician locates the box from which no card has been drawn. How many ways are there to put the cards in the three boxes so that the trick works? (Two ways are considered different if at least one card is put into a different box.)

**FMC**:

```
theorem test :
  let cards : Finset ℕ := Finset.range 100
  let boxes : Finset (Finset ℕ) := {∅, ∅, ∅} --
   Placeholder for actual box definitions
  Finset.card (Finset.filter (fun b : Finset ℕ =>
   Finset.Nonempty b) boxes) = 3 ∧
  (∀ b₁ b₂ : Finset ℕ, b₁ ∈ boxes → b₂ ∈ boxes → b₁
   ≠ b₂ → ∀ x y : ℕ, x ∈ b₁ → y ∈ b₂ →
  ∀ b₃ b₄ : Finset ℕ, b₃ ∈ boxes → b₄ ∈ boxes → b₃ ≠
   b₄ → ∀ z w : ℕ, z ∈ b₃ → w ∈ b₄ →
  x + y ≠ z + w) := by sorry
```

**CombiBench**:

```
abbrev Cards := Finset.Icc 1 100
abbrev Boxes := Fin 3
abbrev Trick := ℕ → Boxes

def trick_works (f : Cards → Boxes) (t : Trick) :
   Prop :=
  ∀ c₁ c₂ : Cards,
  -- given the sum of two cards from box 0 and box 1
   then the trick gives the result of box 2
  (f c₁ = 0 → f c₂ = 1 → t (c₁.1 + c₂.1) = 2) ∧
  -- given the sum of two cards from box 0 and box 2
   then the trick gives the result of box 1
  (f c₁ = 0 → f c₂ = 2 → t (c₁.1 + c₂.1) = 1) ∧
  -- given the sum of two cards from box 1 and box 2
   then the trick gives the result of box 0
  (f c₁ = 1 → f c₂ = 2 → t (c₁.1 + c₂.1) = 0)

theorem imo_2000_p4 (good_allocations : Finset
   (Cards → Boxes))
   (h : ∀ f, f ∈ good_allocations ↔
   Function.Surjective f ∧ ∃ (t : Trick),
   trick_works f t) :
   good_allocations.card = imo_2000_p4_solution :=
   by sorry
```

---

In this case, the formal statement in FMC does not precisely align with the original natural language description. The problem specifies that, given the sum of two cards drawn from two different boxes, the magician can uniquely identify the box from which no card was drawn. This implies that the set of such pairwise sums for each pair of boxes must be disjoint, effectively serving as a signature for the third box.

However, the FMC version fails to explicitly ensure that $(b_1, b_2)$ and $(b_3, b_4)$ refer to two distinct pairs of boxes, violating the original problem constraints. Moreover, the formal statement does not specify the desired conclusion (i.e., the number of valid configurations), resulting in a missing goal. The approach also relies on explicit enumeration rather than set-based abstractions, making the expression unnecessarily verbose.

In contrast, the CombiBench version resolves these issues by introducing explicit box indices and counting mechanisms, leading to a clearer and more faithful formalization. This comparison suggests that, for combinatorial problems, LLM-generated formalizations still fall short, and statements formalized by mathematical experts retain a significant advantage.

## C    APPENDIX: ERROR IN *FMC*

### C.1    ERROR DISTRIBUTION

Table 10: Error distribution in the manually verified FMC subset.

| Error Type | Count |
|---|---|
| Data type error | 1 |
| Condition error | 2 |
| No solution provided | 3 |
| Incorrect proof goal | 2 |
| Modeling failure | 1 |
| Original problem error | 2 |

### C.2    CASE STUDY

#### C.2.1    CASE 1

**Original problem**: Find all positive values of $a$ for which both roots of the equation $a^2x^2 + ax + 1 - 7a^2 = 0$ are integers.

**Formalized output**:

```
theorem test
  (a : ℝ)
  (h₀ : 0 < a)
  (h₁ : ∃ r s : ℤ, a^2 * r ^ 2 + a * r + (1 - 7 * a
    ^ 2) = 0 ∧ a^2 * s ^ 2 + a * s + (1 - 7 * a ^ 2) =
    0) :
  a = 1 ∨ a = 1 / 2 ∨ a = 1 / 3 := by
  sorry
```

**Analysis**: In condition $h_1$, $r$ and $s$ are both solutions of the equation $a^2x^2 + ax + 1 - 7a^2 = 0$, but it is not stated that the equation has exactly these two solutions. In other words, $r$ and $s$ may represent the same solution, while the equation may have another solution that is a non-integer.

### C.2.2 CASE 2

**Original problem**: Let $a_1 < a_2 < \cdots < a_n$ be real numbers. Prove that if $f$ is a bijection mapping $\{a_1, \ldots, a_n\}$ onto itself such that

$$a_1 + f(a_1) < a_2 + f(a_2) < \cdots < a_n + f(a_n),$$

then $f$ is the identity.

**Formalized output**:

```
theorem test
    (n : ℕ)
    (a : Fin n → ℝ)
    (f : Fin n → Fin n)
    (h₀ : StrictMono a)
    (h₁ : Function.Bijective f)
    (h₂ : StrictMono (fun i => a i + a (f i))) :
    f = id := by
    sorry
```

**Analysis**: In the inequality, it should be $f(a_i)$; however, after formalization, it appears as $a(f_i)$. That is, the order of the function and its argument has been swapped, leading to an incorrect indexing and function application.

### C.2.3 CASE 3

**Original problem**: In Geneva there are 16 secret agents, each of whom is watching one or more other agents. It is known that if agent $A$ is watching agent $B$, then $B$ is not watching $A$. Moreover, any 10 agents can be ordered so that the first is watching the second, the second is watching the third, etc, the last is watching the first. Show that any 11 agents can also be so ordered.

**Formalized output**:

```
theorem problem
    (R : Fin 16 → Fin 16 → Prop)
    (h_asymm : ∀ a b, R a b → ¬ R b a)
    (h10 : ∀ (s : Finset (Fin 16)) (hs : s.card = 10),
      ∃ (f : Fin 10 → s), Function.Bijective f ∧
      ∀ i, R (f i) (f (i + 1))) :
    ∀ (t : Finset (Fin 16)) (ht : t.card = 11),
      ∃ (f : Fin 11 → t), Function.Bijective f ∧
      ∀ i, R (f i) (f (i + 1)) := by
    sorry
```

**Analysis**: The last person in the groups of 10 or 11 should monitor the first person. However, in the formalization, it is written as $R(f_i, f_{i+1})$ without applying a modulo operation to the index for the last person. As a result, the indexing is incorrect.

## D APPENDIX: PROMPTS

### D.1 FORMAL TRANSLATION PROMPTS

There are two different stages of formal translation, the first goes with few-shot prompting and the second goes with error feedback. Following is the prompt for translation with few-shot learning.

> A math theorem in natural language will be provided and please translate it into a Lean4 theorem. Please only return the translation (Lean4 code) and no analysis, no mathlib4 import, no comments, no proof, no reasoning. Use ":= by sorry" as a placeholder for proof. Here are some examples for it: {few_shot}. Following the examples above, translate the next problem into Lean4: {problem}

Following is the prompt for translation with error feedback.

> A math theorem in natural language will be provided and please translate it into a Lean4 theorem. Please only return the translation (Lean4 code) and no analysis, no mathlib4 import, no comments, no proof, no reasoning. Use ":= by sorry" as a placeholder for proof. Here is the theorem in natural language: {problem}. Before your translation, note that this problem has been mistranslated as the following. Concrete errors have been listed and please avoid similar mistakes when translating it again. Mistranslation: {failed_info}

### D.2 PROBLEM DECOMPOSITION PROMPTS

Following is the prompt for problem decomposition.

> You are a mathematical semantic analysis assistant. Your task is to analyze a natural language math problem and decompose it into data types, conditions (assumptions), and the proof goal.
>
> Note: If the natural language problem does not explicitly provide the proof goal, calculate the result first and include it as the proof goal.
>
> Return the output strictly in JSON format, without any extra text or explanations. Here are some examples for reference: {NL_shot} Now, this is the natural language math problem: {NL}
>
> Please decompose it and return in the following JSON format only:
> ```
> {"data_type":  ["type1","type2"], "conditions":
> ["condition1","condition2"], "proof_goal":  "Here is
> the goal"}
> ```

### D.3 SEMANTIC ALIGNMENT CHECK PROMPTS

Following is the prompt for semantic alignment check.

> You are a mathematical semantic alignment assistant. You will be given:
> 1. A natural language problem, already decomposed into: {"data_type": [...], "conditions": [...], "proof_goal": "..."}
> 2. A Lean4 problem in its original code.
>
> Your task is to compare the natural language problem decomposition and the Lean4 problem in terms of data_type, conditions and proof_goal, and decide whether they are same or different in their mathematical essence.
>
> Any mismatch in these three area counts result in a "different problem." Note:
>
> Differences in variable names or implicit conditions derived from the given in-

formation do not indicate a different problem. Judge based on mathematical equivalence, not superficial differences.

Return strictly in JSON format: {"Same": true/false, "Analysis": "Briefly explain."}

Now please compare natural language: {NL} and Lean4: {problem_lean}, and decide whether they are same or different in their mathematical essence.

# E APPENDIX: FAILURE OF GEOMETRIC PROBLEMS

---

**Natural Language**

In a triangle $\triangle ABC$, let $m_a, m_b, m_c$ be the lengths of the internal angle bisectors, $M_a, M_b, M_c$ the lengths of these bisectors extended to the circumcircle, and $l_a = \dfrac{m_a}{M_a}$, $l_b = \dfrac{m_b}{M_b}$, $l_c = \dfrac{m_c}{M_c}$.

Prove that $\dfrac{l_a}{\sin^2 A} + \dfrac{l_b}{\sin^2 B} + \dfrac{l_c}{\sin^2 C} \geq 3$ with equality if and only if $\triangle ABC$ is equilateral.

---

**Lean**

```
theorem angle_bisector_inequality
  (A B C : ℝ)
  (hA : 0 < A) (hB : 0 < B) (hC : 0 < C)
  (hA_lt_pi : A < Real.pi) (hB_lt_pi : B < Real.pi) (hC_lt_pi : C < Real.pi)
  (m_a m_b m_c M_a M_b M_c : ℝ)
  (hm_a : 0 < m_a) (hm_b : 0 < m_b) (hm_c : 0 < m_c)
  (hM_a : 0 < M_a) (hM_b : 0 < M_b) (hM_c : 0 < M_c)
  (l_a l_b l_c : ℝ)
  (hl_a : l_a = m_a / M_a) (hl_b : l_b = m_b / M_b) (hl_c : l_c = m_c / M_c) :
  (l_a / (Real.sin A)^2) + (l_b / (Real.sin B)^2) + (l_c / (Real.sin C)^2) ≥ 3 :=
by sorry
```

---

Figure 4: A failed formalization example in geometry.

Figure 4 presents an incorrect formalization example. The original problem, described in natural language, involves a mathematical proposition about the internal angle bisectors of a triangle, where $m_a, m_b, m_c$ denote the lengths of the three angle bisectors, and $M_a, M_b, M_c$ represent their extended lengths intersecting the circumcircle. The objective is to prove an inequality along with the conditions for equality. In the Lean formulation, although the angles $\angle A$, $\angle B$ and $\angle C$ are constrained between $0$ and $\pi$, and all six segment lengths are specified to be positive, with accurate quantitative relationships given for $l_a, l_b, l_c$ relative to the six known segments and the final goal stated, the essential triangle constraint is missing. For example, the condition that the interior angles of $\triangle ABC$ sum to $180°$ is absent, and constraints involving angle bisectors, the circumcircle, and their intersections—although implicit in the original problem—are not imposed. The LLM neglects these implicit constraints during formalization. Additionally, the Lean statement lacks the goal of proving the conditions under which the equality holds.

