# OpenReview forum: "FMC: Formalization of Natural Language Mathematical Competition Problems"
_ICLR.cc/2026/Conference — Submitted to ICLR 2026_

### Official Review · Reviewer_5JX1 · 2025-10-28

**Soundness:** 1
**Presentation:** 2
**Contribution:** 1
**Rating:** 2
**Confidence:** 4

**Summary:**

This paper introduces FMC, an LLM-based autoformalization framework that integrates few-shot prompting, error feedback for syntactic verification, and problem decomposition for semantic alignment. The authors also construct a dataset of 6,994 Lean statements derived from 3,214 Olympiad-level math problems. Experimental results demonstrate that few-shot prompting, error feedback, and decomposition each contribute effectively to improving LLM-based autoformalization performance.

**Strengths:**

* The paper is easy to follow, with a clear motivation.
* The proposed dataset could be valuable for training LLMs in both autoformalization and theorem proving tasks.

**Weaknesses:**

The proposed method lacks novelty. Techniques such as few-shot prompting, syntactic correction through error feedback, and semantic verification have already been explored in prior work [1, 2, 3, 4].

Moreover, the autoformalized statements produced by the system are only guaranteed to be syntactically valid, with semantic consistency checked solely by LLMs. There is no manual verification to ensure that the formal statements truly align with their original natural-language counterparts. In many cases, autoformalization may omit implicit assumptions present in the original problem statements, leading to incorrect or unprovable formulations—despite being syntactically correct. This issue has also been observed in prior work such as DeepSeek Prover, which demonstrates that some automatically formalized statements are actually false (which can be disproved). This could partially explain why existing LLMs achieve relatively low accuracy on the FMC benchmark.

In addition, the baseline comparison is limited to a vanilla LLM, which may only serve as an ablation rather than a true comparison. Stronger and more representative baselines in this area should be considered to provide a fair evaluation.

Finally, the lack of manual validation across the entire dataset raises concerns about the reliability of the proposed benchmark. As shown in Table 4, the automated semantic checking is not highly reliable and shows only weak alignment with human judgments, which further limits the scope and credibility of the dataset.

References:

[1] Autoformalization with Large Language Models

[2] Autoformalize Mathematical Statements by Symbolic Equivalence and Semantic Consistency

[3] Improving Autoformalization using Type Checking

[4] KELPS: A Framework for Verified Multi-Language Autoformalization via Semantic-Syntactic Alignment

**Questions:**

How many samples were used for manual evaluation in Section 5.2?

---

> ### Author Response · Authors · 2025-12-03
> **Rebuttal for Reviewer 5JX1#1**
>
> ### W1: Lacking novelty
>
> Paper [1] presents a traditional approach in which automatic formalization is followed by semantic verification using an LLM-as-a-judge. Paper [2] primarily demonstrates the positive impact of multiple sampling in autoformalization and introduces methods for syntax consistency and semantic consistency, namely equivalence checking among generated statements and comparison between the original problem and informalized statements. Paper [3] employs the Lean compiler for syntax consistency verification, uses few-shot prompting, and incorporates human inspection. The core innovation of paper [4] is the introduction of Knowledge Extraction (KE) as an intermediate step in formalization, enabling the decomposition of mathematical problems.
>
> Indeed, few-shot prompting, syntactic correction via error feedback, and semantic verification are common core techniques. The novelty of our work lies in the **semantic verification** component. We abandon the commonly used LLM-as-a-judge approach and instead adopt a new **decomposing problem** method. Its innovations are as follows:
>
> 1. The decomposing problem method is applied for **semantic verification**, rather than for formalization as in KELPS.
> 2. Compared to KELPS, our decomposing problem approach is **more concise** and **computationally efficient**.
>
> A detailed explanation is provided below:
>
> 1. **Difference in method of analyzing math problems**
>
>    - **KELPS** defines **an intermediate language, knowledge equation (KE)** when splitting the problem, so that it can bridge natural language and formal language. Designing KE requires comprehensive coverage on mathematic concepts and well-defined conversion rules between KE and formal language, which entails **significant preparatory work**.
>
>    - Our work also decomposes problems, but **introduces no intermediate language**. Instead, we focus on **the logical structure** of data types, conditions, and proof goals, allowing LLMs to operate directly and especially helps to **prevent confusion between conditions and conclusions**. This approach is **simpler**, **more lightweight**, and has been **experimentally validated**, aligning with our goal of a training-free, low-cost pipeline.
>
> 2. **Difference in application scenario**
>    - KELPS use this method for **formalization**, while our work applies the process to **semantic checking**. Our experiments show that decomposing problems in this way improves the LLM’s ability to judge semantic consistency. This demonstrates that structured problem decomposition has practical benefits beyond formalization alone.
>
> In summary, the core innovation of our autoformalization pipeline is in the semantic check phase. Rather than relying on a traditional LLM-as-judge approach, we **mimic human reasoning** by decomposing problems along three dimensions: data types, conditions, and proof goals. This helps prevent confusion between conditions and conclusions in formal statements. By combining well-known components, including few-shot prompting, error feedback, and decomposition, and selecting appropriate LLMs, we are able to build a **training-free and practical pipeline**.
>
> ### W2: No manual verification on the whole dataset
>
> Thank you for raising this point. Manual verification is indeed crucial for evaluating dataset quality.
>
> In Table 4, we demonstrate the performance comparison of various LLMs and different semantic alignment methods. The benchmark for these comparisons is derived directly from the results of human inspection. Specifically, **Precision** is defined as the semantic consistent rate for the data selected into the FMC dataset under manual verification, serving as an estimate of the FMC dataset's quality. This metric essentially represents a form of manual verification conducted on a subset of the FMC.
>
> To further confirm our dataset quality, we randomly selected an additional 50 samples as a subset from the FMC and performed a new manual verification, yielding an accuracy result of **78%**. This figure is largely consistent with the initial Precision of 80%. Relative to the dataset's scale, this accuracy rate classifies the data as high-quality.
>
> Regarding **manual validation across the entire dataset**, we must note that this approach is **impractical for large-scale datasets** due to the significant cost of human inspection. Many large datasets, including Proverbench which you mentioned, are known to contain a certain amount of inevitable error. Given the **scale and difficulty** of our dataset, the 78% accuracy rate is acceptable. We acknowledge that neglecting implicit conditions can potentially lead to unprovable propositions, but our data indicates that the proportion of errors caused by incorrect conditions is relatively low.

---

> > ### Author Response · Authors · 2025-12-03
> > **Rebuttal for Reviewer 5JX1#2**
> >
> > **experiments**
> >
> > The statistics on error types found in the 50 sampled entries are provided below.
> >
> > | Error Type             | Count |
> > | ---------------------- | ----- |
> > | Data type error        | 1     |
> > | Condition error        | 2     |
> > | No solution provided   | 3     |
> > | Incorrect proof goal   | 2     |
> > | Modeling failure       | 1     |
> > | Original problem error | 2     |
> >
> > *Some natural language problems are wrong in the first place, due to extraction error.
> >
> > **Case study**
> >
> > *[case 1]*
> >
> > - Natural language:
> >
> >   "Find all positive values of $a$ for which both roots of the equation $a^{2} x^{2}+a x+1-$ $7 a^{2}=0$ are integers."
> >
> > - Formal code:
> >
> >   ```
> >   theorem test
> >     (a : ℝ)
> >     (h₀ : 0 < a)
> >     (h₁ : ∃ r s : ℤ, a^2 * r ^ 2 + a * r + (1 - 7 * a ^ 2) = 0 ∧ a^2 * s ^ 2 + a * s + (1 - 7 * a ^ 2) = 0) :
> >     a = 1 ∨ a = 1 / 2 ∨ a = 1 / 3 := by
> >     sorry
> >   ```
> >
> > - Explanation：
> >
> >   In condition $h_1$, $r$ and $s$ are both solutions of the equation $a^{2} x^{2} + a x + 1 - 7a^{2} = 0$, but it is not stated that the equation has exactly these two solutions. In other words, $r$ and $s$ may represent the same solution, while the equation may have another solution that is a non-integer.
> >
> > *[case 2]*
> >
> > - Natural language:
> >
> >   "Let \(a_{1} < a_{2} < \cdots < a_{n}\) be real numbers. Prove that if \(f\) is a bijection mapping \(\{a_{1}, \ldots, a_{n}\}\) onto itself such that
> >
> >   $$
> >   a_{1} + f(a_{1}) < a_{2} + f(a_{2}) < \cdots < a_{n} + f(a_{n}),
> >   $$
> >
> >   then \(f\) is the identity."
> >
> > - Formal code:
> >
> >   ```
> >   theorem test
> >     (n : ℕ)
> >     (a : Fin n → ℝ)
> >     (f : Fin n → Fin n)
> >     (h₀ : StrictMono a)
> >     (h₁ : Function.Bijective f)
> >     (h₂ : StrictMono (fun i => a i + a (f i))) :
> >     f = id := by
> >     sorry
> >   ```
> >
> > - Explanation：
> >
> >   In the inequality, it should be $f(a_i)$; however, after formalization, it appears as $a(f_i)$. That is, the order of the function and its argument has been swapped, leading to an incorrect indexing and function application.
> >
> > *[case 3]*
> >
> > - Natural language:
> >
> >   "In Geneva there are 16 secret agents, each of whom is watching one or more other agents. It is known that if agent $A$ is watching agent $B$, then $B$ is not watching $A$. Moreover, any 10 agents can be ordered so that the first is watching the second, the second is watching the third, etc, the last is watching the first. Show that any 11 agents can also be so ordered."
> >
> > - Formal code:
> >
> >   ```
> >   theorem problem
> >     (R : Fin 16 → Fin 16 → Prop)
> >     (h_asymm : ∀ a b, R a b → ¬ R b a)
> >     (h10 : ∀ (s : Finset (Fin 16)) (hs : s.card = 10),
> >       ∃ (f : Fin 10 → s), Function.Bijective f ∧
> >       ∀ i, R (f i) (f (i + 1))) :
> >     ∀ (t : Finset (Fin 16)) (ht : t.card = 11),
> >       ∃ (f : Fin 11 → t), Function.Bijective f ∧
> >       ∀ i, R (f i) (f (i + 1)) := by
> >     sorry
> >   ```
> >
> > - Explanation：
> >
> >   The last person in the groups of 10 or 11 should monitor the first person. However, in the formalization, it is written as $R(f_i, f_{i+1})$ without applying a modulo operation to the index for the last person. As a result, the indexing is incorrect.
> >
> > **Action**: We will incorporate this content into the main text, and case study will be placed in the appendix.
> >
> > ### W3: Suggestion for stronger and more representative baseline comparison
> >
> > We thank the reviewer for the suggestion regarding baseline selection. Our work employs vanilla LLMs within a task-specific autoformalization pipeline, which incorporates iterative error feedback and problem decomposition mechanisms. In our experiments, we evaluated multiple vanilla LLMs under this pipeline, as using vanilla LLMs as autoformalization tools is a **common and reasonable choice**. Therefore, this was not a deliberate selection of a weaker model. In contrast, KELPS employs a pipeline different from ours, and FormalMath adopts advanced methods that combine state-of-the-art models with an agentic framework. However, most of these approaches are **not publicly available**, making precise reproduction difficult and limiting their use as baselines for comparison.

---

> > > ### Author Response · Authors · 2025-12-03
> > > **Rebuttal for Reviewer 5JX1#3**
> > >
> > > ### W4: Automated semantic check making FMC unreliable
> > >
> > > In this experiment, we conducted manual verification on a total of **39 problems**. This corresponds to one complete run of our pipeline, in which we started from 100 original instances, removed geometry-related problems, and applied Deepseek-R1 for formalization, counting all problems that passed the formalization check. Since Table 4 already clearly demonstrates the performance comparison of various LLMs and different semantic alignment methods, with noticeable differences, we did not perform evaluation on a larger dataset.
> > >
> > > However, as you previously suggested, to further demonstrate the quality of the FMC dataset, we conducted additional manual evaluation. We randomly sampled 50 instances from FMC and performed verification, obtaining an accuracy of **78%**. This result is largely consistent with the **Precision of 80%** reported in Table 4. Considering the size of the dataset, this indicates that FMC is of relatively high quality.

---

### Official Review · Reviewer_3eLr · 2025-10-31

**Soundness:** 3
**Presentation:** 2
**Contribution:** 2
**Rating:** 4
**Confidence:** 3

**Summary:**

This paper introduces an autoformalization pipeline that integrates problem decomposition, few-shot learning, and error feedback to translate natural language mathematical competition problems into Lean 4 formalizations. Using this approach, the authors build FMC, an Olympiad-level dataset aligning natural language problems with their formal Lean counterparts. The FMC dataset contains 3,214 problems and 6,994 corresponding Lean statements. Experimental results highlight its difficulty and demonstrate its potential as a valuable benchmark for advancing research in formal reasoning.

**Strengths:**

- The overall idea of the paper is well-motivated and technically sound. The combination of few-shot learning and error feedback effectively enriches contextual information, which can improve the robustness and accuracy of the autoformalization process. Moreover, problem decomposition bridges the gap between natural language expressions and their formal counterparts in Lean, facilitating more reliable semantic alignment and verification.
- The paper conducts a thorough and systematic ablation study. The experiments clearly isolate and quantify the contributions of each component—problem decomposition, few-shot learning, and error feedback—providing strong empirical evidence for the effectiveness of the proposed pipeline.
- The work also offers practical value beyond methodology. By producing a large-scale Olympiad-level dataset (FMC), it contributes a challenging and well-structured benchmark for evaluating formal reasoning systems and automated theorem provers.

**Weaknesses:**

- The paper contains several presentation inconsistencies that detract from readability and professionalism. For instance, Table 4 is never cited in the main text, and the captions for some figures and tables are overly minimal. Captions such as that of Figure 1 should include a more detailed explanation of the depicted pipeline. Additionally, there is a numerical inconsistency in Section 4.3: the values 3,922 and 87.14% appear only once, whereas 3,214 and 66.99% are used elsewhere. This discrepancy should be clarified and corrected.
- The workflow for this method lacks sufficient detail. Conceptually, it involves two distinct semantic judgments:
(a) verifying whether the decomposed subproblems remain faithful to the original natural language statement, and
(b) checking whether these decomposed elements semantically correspond to the resulting Lean 4 formalization.
The current text does not clearly distinguish these steps, leaving the reader uncertain about the precise evaluation process. Section 5.2 should be expanded to provide a clearer methodological description and include supporting experimental evidence.
- The paper would benefit from a dedicated section describing the dataset itself: its composition, domain coverage (e.g., algebra, number theory, combinatorics), and its comparison with existing datasets. Although comparisons are provided in the appendix, they are never referenced or summarized in the main body, which weakens the paper’s contribution as a benchmark dataset.
- The appendix contains valuable analyses, including dataset comparisons and case studies, but these are not mentioned or cited in the main paper. This omission results in a structural disconnect, rendering the supplementary material underutilized. Integrating references to the appendix within the main text would improve coherence and readability.

**Questions:**

1. The discussion of prior autoformalization research appears rather limited, referencing only [1]. Several important related works are missing; please refer to the recent surveys [2, 3] for a broader overview. In addition, comparative experiments with other recent autoformalization pipelines and tools would strengthen the paper by clarifying how the proposed approach differs from and improves upon existing frameworks.

2. The paper states that state-of-the-art provers include DeepSeek-Prover-V2 and Goedel-Prover-V2, yet the experiments rely on DeepSeek-Prover-V1.5-RL and Goedel-Prover. Could the authors elaborate on the rationale behind this choice? For completeness, it would also be valuable to include STP, which is cited in the paper and represents another leading system. Incorporating results from these newer models would provide a more comprehensive and up-to-date evaluation of the proposed FMC benchmark.

[1] Autoformalization with large language models, NeurIPS 2022.

[2] A Survey on Deep Learning for Theorem Proving, COLM 2024.

[3] Autoformalization in the Era of Large Language Models: A Survey, arXiv 2025.

---

> ### Author Response · Authors · 2025-12-03
> **Rebuttal for Reviewer 3eLr#1**
>
> We thank the reviewers for their careful reading and for recognizing **the soundness of our pipeline**, **thoroughness of ablation experiments**, and the value of the proposed datasets as **a valuable benchmark for future research**. We agree that the concerns raised are valuable for improving the clarity of our presentation. Below we respond to each point and aim to address your concerns.
>
> ### W1: Presentation inconsistencies and captions of figure
>
> We appreciate your careful review and for pointing out the errors in our manuscript, particularly the data inconsistency in Section 4.3. The discrepancies likely arose from a significant update during the development of our work, during which some data may have been accidentally overlooked and remained unchanged. The earlier version of the dataset contained 3,922 entries, and since the semantic check method had not been improved at that time, it led to an overestimated formalization accuracy of 87.14%. This issue has now been corrected in the revised paper. Other concerns, such as inconsistencies in wording, table references, and figure captions, have also been addressed in the updated version.
>
> ### W2: Lack sufficient detail of semantic judgments
>
> Thank you for raising this question. It seems that the semantic consistency check method in our work has caused some confusion. Here, we clarify the workflow and address your concerns.
>
> 1. **Decomposition as a method for semantic checking**
>
>    In this work, decomposing problems serves as an intermediate approach to determine whether the semantics of the original natural language problem and the formalized problem are consistent, and it is **not applied to the formalization process** itself. Specifically, to check semantic consistency between the original natural language problem and the formal statement, we first decompose the natural language problem and then compare the decomposed problem with the formal statements for consistency.
>
> 2. **No additional checks for the issues in (a)**
>
>    The decomposition process is essentially a logical breakdown that transforms plain text data into semi-structured data. The specific descriptions of “data_type”, “conditions”, and “proof_goal” remain primarily in natural language. Since **problem decomposition** does not involve translation from natural language to Lean, it is relatively straightforward and **rarely causes semantic inconsistencies**. Therefore, we do not perform additional semantic checks for the issues mentioned in (a); instead, we use the semantic consistency between the formal problems and the decomposed problems as a proxy for consistency with the original problem. Table 4 in Section 5.2 presents a comparison of these two consistency-checking methods.
>
> 3. **Advantages of using problem decomposition for semantic consistency checking**
>
>    Compared to previous methods, such as Lean Workbook, which first **back-translates** formal problems into natural language and then performs semantic checks within the natural language, or the traditional **“LLM-as-a-judge”** approach that directly compares formal problems with the original natural language problems, we find that applying problem decomposition in the semantic check phase offers **clear advantages**. By mimicking the way humans formalize problems, we decompose problems into three aspects—data type, conditions, and proof_goal—aiming to **avoid confusion between “conditions” and “conclusions”** in the formalized statements.

---

> > ### Author Response · Authors · 2025-12-03
> > **Rebuttal for Reviewer 3eLr#2**
> >
> > ### W3: No dataset overview and comparison with other dataset not mentioned in the main body
> >
> > Thank you for your suggestion. We have added a detailed description of the characteristics of our dataset and included a comparison with existing datasets. The following table reports the distribution of our FMC dataset across mathematical domains and provides a comparison with FormalMath and Lean Workbook. The corresponding figure will be included in the appendix.
> >
> > **Table 1. FMC Dataset Distribution**
> >
> > | Domain        | Count | Percent |
> > | ------------- | ----- | ------- |
> > | Algebra       | 380   | 11.8%   |
> > | Analysis      | 977   | 30.4%   |
> > | Number Theory | 1100  | 34.2%   |
> > | Combinatorics | 466   | 14.5%   |
> > | Probability   | 21    | 0.7%    |
> > | Set Theory    | 16    | 0.5%    |
> > | Other         | 254   | 7.9%    |
> >
> > **Table 2. FormalMath Dataset Distribution**
> >
> > | Domain               | Count | Percent |
> > | -------------------- | ----- | ------- |
> > | Algebra              | 2629  | 47.3%   |
> > | Number Theory        | 1135  | 20.4%   |
> > | Discrete Mathematics | 438   | 7.9%    |
> > | Applied Mathematics  | 458   | 8.2%    |
> > | Geometry             | 295   | 5.3%    |
> > | Precalculus          | 186   | 3.3%    |
> > | Calculus & Advanced  | 359   | 6.4%    |
> > | Other                | 60    | 1.1%    |
> >
> > **Table 3. Lean Workbook Dataset Distribution**
> >
> > | Category      | Count | Percent |
> > | ------------- | ----- | ------- |
> > | inequality    | 46847 | 36.03%  |
> > | Algebra       | 45218 | 34.77%  |
> > | Number Theory | 22813 | 17.55%  |
> > | Analysis      | 2799  | 2.15%   |
> > | Combinatorics | 893   | 0.69%   |
> > | trigonometry  | 4133  | 3.18%   |
> > | equation      | 3255  | 2.50%   |
> > | proof         | 3172  | 2.44%   |
> > | other         | 905   | 0.70%   |
> >
> > **Notes**: Please note that each dataset follows its own publicly defined categorization scheme, and these schemes differ across datasets. For clearer comparison, we merged several categories where appropriate; however, some datasets inherently exhibit a richer variety of labels. It is important to emphasize that the diversity in category distributions arises not only from the dataset content itself, but also from differences in their underlying classification standards.
> >
> > **Analysis**: As shown, the FMC dataset demonstrates a relatively balanced and diverse distribution across mathematical domains.
> >
> > **Action**: We will incorporate this content into the appendix, and a corresponding reference will added in the main text.
> >
> > ### W4: Appendix not mentioned in the main body
> >
> > Your suggestions are highly valuable. We will revise the paper accordingly and include appropriate references to the relevant appendix sections, aiming to further improve the coherence and readability of the manuscript.
> >
> > ### Q1: Insufficient references to survey papers
> >
> > Thank you for your suggestions. We have **already cited the first paper**, and we agree that the other two surveys you mentioned are also highly valuable references. Survey [2] provides a rich and comprehensive overview of **theorem proving**, especially formal-language-based theorem proving. Survey [3] offers an in-depth investigation of **autoformalization scenarios and workflows**, making it particularly relevant to our work.
> >
> > Due to space limitations, we did not discuss or cite these two surveys in the main text, especially survey [2], which focuses on **theorem proving** rather than autoformalization. Nevertheless, we were indeed aware of important works highlighted in survey [3], such as Lean Workbook, which also mentioned in the paper. We have now added a citation to survey [3] in the revised version.

---

> > > ### Author Response · Authors · 2025-12-03
> > > **Rebuttal for Reviewer 3eLr#3**
> > >
> > > ### Q2:  More up-to-date evaluation on FMC benchmark
> > >
> > > Thank you very much for your suggestion. As this work was completed relatively early, there were some oversights in the evaluation. We now supplement the evaluation results for **DeepSeek-Prover-V2-671B** and **Goedel-Prover-V2**. Experiments were conducted by randomly sampling 1,000 instances from FMC, with 32 samples per instance. The results are summarized in the table below. All unmarked entries correspond to 32 samples per instance.
> > >
> > > | Model                       | FMC   | miniF2F          | Proverbench | FormalMath | FormalMath-Lite | ProofNet         |
> > > | --------------------------- | ----- | ---------------- | ----------- | ---------- | --------------- | ---------------- |
> > > | **DeepSeek-Prover-V2-671B** | 40.5% | 82.4%            | 49.5%       | 28.31%     | 56.00%          | 23.8%            |
> > > | **Goedel-Prover-V2**        | 62.5% | 88.1%            | –           | –          | –               | –                |
> > > | **STP**                     | –     | 61.2% (pass@128) | –           | 13.87%     | 48.59%          | 19.5% (pass@128) |
> > >
> > > **Analysis**: From the results, it can be observed that FMC remains substantially **more challenging** compared to miniF2F, ProverBench, and FormalMath-Lite. Moreover, the strong performance of Goedel-Prover-V2 on FMC further confirms the **high quality of formalization**, with no significant number of erroneous statements.
> > >
> > > **Notes**: Regarding STP, it is both computationally expensive and no longer state-of-the-art in theorem proving. On benchmarks such as ProofNet, FormalMath, and FormalMath-Lite, its performance is inferior to that of DeepSeek-Prover-V2-671B (no public results are available for Goedel-Prover-V2). Even under @128 sampling, its performance on miniF2F remains below that of DeepSeek-Prover-V2-671B and Goedel-Prover-V2. Therefore, we did not perform related experiments on FMC.
> > >
> > > **Action**: We will incorporate this content into the paper, and evaluate more SOTA models when new models are released.

---

### Official Review · Reviewer_SWQd · 2025-11-01

**Soundness:** 2
**Presentation:** 2
**Contribution:** 2
**Rating:** 4
**Confidence:** 3

**Summary:**

This paper presents FMC, an autoformalization pipeline that translates natural-language mathematical competition problems (mainly Olympiad-level) into the Lean formal language. The proposed approach integrates LLMs with error feedback and problem decomposition to achieve a fully automatic, training-free formalization process. Using this pipeline, the authors construct a dataset of 3,214 natural-language problems and 6,994 corresponding Lean statements, achieving 93.4% syntactic validity and 66.9% semantic consistency, both surpassing prior work (e.g., StepFun-Formalizer). The paper further evaluates multiple LLMs (DeepSeek-R1, GPT-4o-mini, Claude 3.7 Sonnet) and several automated theorem provers on the dataset, showing FMC’s challenging nature and its potential as a benchmark for formal reasoning.

**Strengths:**

- The proposed combination of error feedback and problem decomposition provides a well-motivated and empirically validated improvement over prior autoformalization frameworks. The training-free design is elegant and practical.
- The dataset focuses on Olympiad-level problems, ensuring both semantic richness and difficulty. The reported statistics (93% syntactic, 67% semantic accuracy) are impressive given the full automation and task complexity.
- The paper conducts detailed comparisons across multiple models (DeepSeek, GPT-4, Claude) and theorem provers (Kimina, Goedel, DeepSeek-Prover). This breadth of evaluation makes FMC a valuable benchmark for future research in formal mathematical reasoning.

**Weaknesses:**

- While the combination of known components (few-shot prompting, error feedback, decomposition) works well, the conceptual novelty is somewhat incremental compared to existing autoformalization frameworks such as StepFun-Formalizer or KELPS.
- The paper includes some case studies, but a deeper quantitative or typological analysis of the 33% semantic failures would strengthen the understanding of model limitations and inform future improvements.
- The paper does not specify whether the dataset, code, or prompts will be released. Given its claimed benchmark status, public release (and clear licensing details) would be essential for meaningful community adoption.

**Questions:**

The proposed pipeline focuses on Olympiad-level problems with relatively well-structured text.

1. How well does FMC generalize to non-Olympiad mathematical texts (e.g., research papers, textbooks)?

2. Would the error feedback and decomposition mechanisms still work effectively on less formulaic or multi-step narrative problems?

The paper highlights that syntax error feedback improves formalization accuracy, while semantic error feedback may introduce noise.

1. Could the authors elaborate on the computational cost and iteration depth of the feedback loop?
2. How many rounds of retranslation were typically needed before convergence, and how does this trade off with the observed accuracy gains?
3. Have the authors explored adaptive stopping criteria or selective feedback strategies to balance quality and efficiency?

---

> ### Author Response · Authors · 2025-12-03
> **Rebuttal for Reviewer SWQd#1**
>
> We thank the reviewers for their careful reading and for recognizing our work’s **improvement over prior autoformalization frameworks**, **training-free design**, and the value of the proposed datasets as **a valuable benchmark for future research**. We agree that the concerns raised are valuable for improving the clarity of our presentation. Below we respond to each point and aim to address your concerns.
>
> ### W1: Concerns about novelty and difference from StepFun-Formalizer or KELPS
>
> We appreciate your careful reading, which has made us aware that the novelty of our work may be subject to misinterpretation. Here, we would like to clarify the main contributions and innovations of this work.
>
> In fact, our main innovation lies in the **FMC dataset**, which is both **large-scale and highly challenging** comparing to other datasets. It is an Olympiad-level dataset aligning natural language problems with Lean formalizations, containing 3,214 natural language mathematical problems and 6,994 corresponding Lean statements. The formalization pipeline, however, serves as **a secondary innovation** to support the construction of this dataset. And, as you have noticed, in the construction of autoformalization pipeline, previous studies have also explored semantic connections between natural language and the formal language Lean. But our work differs in several important aspects.
>
> 1. **Difference in method of analyzing math problems**
>
>    - **StepFun-Formalizer**, for example, performs **unstructured analysis**, presenting reasoning in natural language. Also, it focuses on how to understand each **mathematical concept** and translate it into Lean.
>
>    - **KELPS**, on the other hand, applies structured decomposition. It splits problems into 'declaration', 'fact' and 'query'. KELPS also defines **an intermediate language, knowledge equation (KE)**, to bridge natural language and formal language. Designing KE requires comprehensive coverage on mathematic concepts and well-defined conversion rules between KE and formal language, which entails **significant preparatory work**.
>
>    - Our work also **decomposes problems into structures**, but **introduces no intermediate language**. Instead, we focus on **the logical structure** of data types, conditions, and proof goals, allowing LLMs to operate directly and especially helps to **prevent confusion between conditions and conclusions**. This approach is **simpler**, **more lightweight**, and has been **experimentally validated**, aligning with our goal of a training-free, low-cost pipeline.
>
> 2. **Difference in application scenario**
>    - StepFun-Formalizer and KELPS use their methods for **formalization**, while our work applies the process to **semantic checking**. Our experiments show that decomposing problems in this way improves the LLM’s ability to judge semantic consistency. This demonstrates that structured problem decomposition has practical benefits beyond formalization alone.
>
> In summary, the core innovation of our autoformalization pipeline is in the semantic check phase. Rather than relying on a traditional LLM-as-judge approach, we **mimic human reasoning** by decomposing problems along three dimensions: data types, conditions, and proof goals. This helps prevent confusion between conditions and conclusions in formal statements. By combining well-known components, including few-shot prompting, error feedback, and decomposition, and selecting appropriate LLMs, we are able to build a **training-free and practical pipeline**.
>
> **Action**: We will briefly include this part in the paper to clarify our novelty, helping readers better understand the contributions of our work.

---

> > ### Author Response · Authors · 2025-12-03
> > **Rebuttal for Reviewer SWQd#2**
> >
> > ### W2: Lack of quantitative analysis of semantic failures
> >
> > We appreciate your observation. We agree that, in addition to detailed case studies, a **statistical analysis** of semantic failures is also valuable for understanding the model's limitations. Therefore, we randomly selected 50 problems that failed the semantic consistency check and categorized the types of errors. The results are summarized in the table below.
> >
> > | Problem Type      | Count | Secondary Problem Type            | Count |
> > | ----------------- | ----- | --------------------------------- | ----- |
> > | Data type error   | 3     | Incorrect data type               | 3     |
> > | Condition error   | 14    | Condition too strong or too weak  | 2     |
> > |                   |       | Missing constraint                | 6     |
> > |                   |       | Incorrect constraint              | 6     |
> > | Proof goal error  | 26    | No solution provided              | 13    |
> > |                   |       | Incomplete solution               | 4     |
> > |                   |       | Incorrect solution                | 3     |
> > |                   |       | Incorrect proof goal              | 2     |
> > |                   |       | Incomplete proof goal             | 1     |
> > |                   |       | Direct output of few-shot example | 3     |
> > | Modeling failure  | 4     | --                                | 4     |
> > | * Judgement error | 7     | --                                | 7     |
> >
> > *Judgement error indicates that target statement is mistaken for semantic failure.
> >
> > **The sum of count is more than 50 because some problems have several mistakes in one.
> >
> >
> >
> > We further examined their **distribution across mathematical topics**. Problems without a solution are often related to finding functions or determining value ranges, while modeling failures typically occur in applied problems and combinatorics. Additionally, questions involving maxima, minima, or attainable function values are frequently overlooked during formalization. The distribution of semantic failures across mathematical domains is summarized in the table below.
> >
> > | Category      | Count | Percentage |
> > | ------------- | ----- | ---------- |
> > | Algebra       | 21    | 21.0%      |
> > | Analysis      | 24    | 24.0%      |
> > | Number Theory | 26    | 26.0%      |
> > | Combinatorics | 20    | 20.0%      |
> > | Probability   | 1     | 1.0%       |
> > | Set Theory    | 1     | 1.0%       |
> > | Other         | 7     | 7.0%       |
> >
> > **Analysis**: As shown in the table, semantic problems occur most frequently in proof goal error, followed by condition error. For proof goal problems, many problems without a solution were either not solved or incorrectly solved, which remains one of the current challenges.
> >
> > **Action**: We will include this data in the appendix, and future work will further investigate the formalization of problems without solutions.
> >
> > ### W3: No dataset, code or prompts released
> >
> > We appreciate your suggestion. The prompts used in this work have already been provided in the appendix. The full code and dataset will also be made publicly available upon acceptance of the paper, under a permissive license, to facilitate community access and use.

---

> > > ### Author Response · Authors · 2025-12-03
> > > **Rebuttal for Reviewer SWQd#3**
> > >
> > > ### Q1: Performance of proposed pipeline on less-structured text
> > >
> > > We appreciate your suggestion. Following your recommendation, we randomly selected 100 examples each from **ProofNet** and **GSM8K** for additional experiments. The results are presented below, followed by a detailed explanation of the experimental setup and an analysis of the outcomes.
> > >
> > > |          | Input Number | syntax pass | nli_pass |
> > > | -------- | ------------ | ----------- | -------- |
> > > | ProofNet | 100          | 78          | 73       |
> > > | GSM8K    | 100          | 100         | 100      |
> > >
> > > 1. **Non-Olympiad mathematical texts**
> > >
> > >    **ProofNet** is a benchmark for autoformalization and formal proving of **undergraduate-level** mathematics. The problems are primarily drawn from popular **undergraduate pure mathematics textbooks**. We randomly sampled 15 examples from the 73 problems that passed the checks and found one formalization error, resulting in a validity rate of 93.33% within the dataset. For the case study, we selected one example each from the automatically formalized incorrect and correct instances.
> > >
> > >    - **Wrong Case Study**
> > >
> > >      - Natural language:
> > >
> > >        "Prove that $$
> > >        |\mathbf{x}+\mathbf{y}|^{2}+|\mathbf{x}-\mathbf{y}|^{2}=2|\mathbf{x}|^{2}+2|\mathbf{y}|^{2}$$ if $\mathbf{x} \in R^{k}$ and $\mathbf{y} \in R^{k}$."
> > >
> > >      - Formal code:
> > >
> > >        ```
> > >        theorem test
> > >          (k : ℕ)
> > >          (x y : Fin k → ℝ) :
> > >          ‖x + y‖^2 + ‖x - y‖^2 = 2 * ‖x‖^2 + 2 * ‖y‖^2 := by sorry
> > >        ```
> > >
> > >      - Explanation：
> > >
> > >        In Mathlib, the type Fin k → ℝ carries the sup norm (ℓ∞ norm) by default, To ensure the Euclidean (ℓ²) norm, EuclideanSpace ℝ (Fin k) should be used.
> > >
> > >    - **Correct Case Study**
> > >
> > >      - Natural language:
> > >
> > >        "If $A$ and $B$ are disjoint closed sets in some metric space $X$, prove that they are separated."
> > >
> > >      - Formal code:
> > >
> > >        ```
> > >        theorem test
> > >          {X : Type*} [MetricSpace X]
> > >          (A B : Set X)
> > >          (h_disj : Disjoint A B)
> > >          (hA : IsClosed A)
> > >          (hB : IsClosed B) :
> > >          (closure A ∩ B = ∅) ∧ (A ∩ closure B = ∅) := by sorry
> > >        ```
> > >
> > >      - Explanation：
> > >
> > >        The formalization correctly defines separated sets in a metric space as each being disjoint from the other's closure, which holds for disjoint closed sets. The conditions are transformed correctly as well.
> > >
> > > 2. **Less formulaic or multi-step narrative problems**
> > >
> > >    **GSM8K** (Grade School Math 8K) is a dataset of linguistically diverse grade school math word problems. The dataset was created to support the task of question answering on basic mathematical problems that **require multi-step reasoning**. We randomly sampled 15 examples from the 100 problems that passed the checks and did not find any errors, resulting in a 100% validity rate within the dataset. For the case study, we randomly selected one example.
> > >
> > >    - **Correct Case Study**
> > >
> > >      - Natural language:
> > >
> > >        "Sam works at the Widget Factory, assembling Widgets. He can assemble 1 widget every 10 minutes. Jack from the loading dock can help assemble widgets when he doesn't have anything else to do. When he helps, they put together 2 complete widgets every 15 minutes. Recently the factory hired Tony to help assemble widgets. Being new to the job, he doesn't work as fast as Sam or Jack. Yesterday Sam worked for 6 hours before he had to leave work early for a dentist appointment. Jack was able to help out for 4 hours before he had to go back to the loading dock to unload a new shipment of widget materials. Tony worked the entire 8-hour shift. At the end of the day, they had completed 68 widgets. How long does it take Tony to assemble a Widget, in minutes?"
> > >
> > >      - Formal code:
> > >
> > >        ```
> > >        theorem test
> > >          (t : ℝ)
> > >          (h_t_pos : t > 0)
> > >          (h_work : 360 * (1/10 : ℝ) + 240 * ((2/15 : ℝ) -(1/10 : ℝ)) + 480 / t = 68) :
> > >          t = 20 := by sorry
> > >        ```
> > >
> > >      - Explanation：
> > >
> > >        The formalization correctly converted between hours and minutes, established the equality relating the total workload to the sum of each individual's workload, and finally listed the equation correctly.
> > >
> > > These results demonstrate that our pipeline exhibits a relatively high formalization capability for textbook problems and multi-step problems.

---

> > > > ### Author Response · Authors · 2025-12-03
> > > > **Rebuttal for Reviewer SWQd#4**
> > > >
> > > > ### Q2: Iteration depth of the feedback loop
> > > >
> > > > To more clearly address your question, we randomly selected 200 examples from the data source. After the first round of formalization, these examples underwent three iterations. As stated in the paper, syntax error feedback improves formalization accuracy, whereas semantic error feedback may introduce noise. In our experiments, we focus solely on syntax error feedback. The experimental results are summarized in the table.
> > > >
> > > > |         | Input Number | Syntax Pass | Syntax Pass/Input Number | Syntax Fail | NLI Pass | Tokens     | Tokens/NLI Pass (cumulative) |
> > > > | ------- | ------------ | ----------- | ------------------------ | ----------- | -------- | ---------- | ---------------------------- |
> > > > | 0-iter. | 200          | 173         | 86.5%                    | 27          | 93       | 37,618,994 | 404,651                      |
> > > > | 1-iter. | 27           | 15          | 55.6%                    | 12          | 10       | 1,276,889  | 377,566                      |
> > > > | 2-iter. | 12           | 3           | 25.0%                    | 9           | 3        | 310,778    | 369,870                      |
> > > > | 3-iter. | 9            | 2           | 22.2%                    | 7           | 0        | 250,130    | 372,239                      |
> > > >
> > > > 1. The table presents the computational cost and iteration depth of the feedback loop. The last column shows the ratio of cumulative tokens to cumulative NLI passes. It can be observed that in the third iteration, the average token cost per NLI pass increases. This is because the number of instances passing both the syntax and semantic checks decreases. However, the token count in each iteration is influenced by multiple factors, including the number of inputs and the syntax pass rate, while the NLI passes are affected by the success rate of semantic checks. Therefore, the trend in average token cost per NLI pass should be interpreted with caution.
> > > >
> > > > 2. Typically, two rounds of retranslation were required before the syntax pass rate stabilized, and by the third iteration, the number of NLI passes no longer increased. In the first iteration, the syntax pass rate remained relatively high, but it dropped sharply in the second iteration. It should be noted that the ratio of NLI passes to syntax passes is not directly determined by the syntax pass rate, but is influenced by the nature of the problems themselves and is thus somewhat stochastic. For efficiency reasons, we only performed a single iteration, maintaining a syntax pass rate of approximately 55% to prevent extremely low NLI pass rates and unnecessary resource consumption in subsequent stages.
> > > >
> > > > 3. Our current work does not adopt adaptive stopping criteria or selective feedback strategies. We acknowledge that these suggestions are reasonable, and we may explore these approaches in future work.

---

### Meta-Review · Area_Chair_MoK8 · 2026-01-06

**Summary:**

The paper is overall not good enough. First, I find the "story" of the paper confusing and weak. In particular, while during the rebuttal the authors try to claim the main contribution of the paper is the size and complexity of the benchmark, the paper only says in one place that this is **one** of its goals. The rest of the paper, including crucially the experimental and method sections, rarely justify the authors' technical choices and never link them to the "bigger picture" of the paper. This also makes judging the paper on its merits hard because those merits in the current version are not emphasized, and are not well compared to prior work. This also results in a weak and confusing experimental section where comparison to prior baselines doesn't exist, and instead, only ablations, some of which are seemingly unimportant (as they are not linked to the story of the paper), are the experiments that are being emphasized there. On top of that, the paper does a poor job of documenting and explaining its experimental setup, metrics, results, and methods, and it is riddled with bad writing, such as the reported wrong number in the experimental section, and the experiment mentioned in the introduction that is not supplied in the paper. Finally, but also very crucially, the authors only promise but do not supply their benchmark to the reviewers. Benchmarking papers in AI for math crucially require their benchmarks to be made available during the review, as many poor-quality benchmarks have been proposed in the community. This is coupled with the fact that the authors do not provide enough information about the human evaluation of the 100 samples they analyzed manually, further making their results inherently unverifiable.

**Reviewer Concerns:**

**Outstanding Reviewer concerns**
- **Positioning and Experimental Comparison to Prior Work (Reviewer SWQd, Weakness 1, Reviewer 3eLr Question 1, Reviewer 5JX1 Weakness 1 and 3)**
While the authors have somewhat answered the positioning question (only in the rebuttal, not the paper), I still find myself unconvinced of the technical novelty of the paper. Part of the issue is that the authors, both in the technical part and the experimental one, fail to disambiguate the paper's contributions and things like syntax feedback that are well known to help autoformalization. To be clear, I am not arguing against the syntax feedback experiment, but when the authors treat it as part of their framework and an experiment with equivalent importance, it confuses the reader about how to judge the paper's contribution.  Another part of the issue is that, indeed, technically, the paper is not that deep.
The issue is made a lot worse, however, by the fact that outside of the NLI experiment in Table 4, there is **no comparison with any auto-formalization** method. Note that the NLI experiment itself is also not an end-to-end experiment against any prior work. This alone is major enough to justify rejection.
Finally, while I will discuss at greater length later, I wanted to point out that on L60 of the paper, the authors mention a comparison to StepFun, yielding 40.5% semantic consistency. This is an experiment I cannot locate in the paper (neither by searching StepFun, nor by searching for this number). This contributes to the idea that a baseline experiment needs to be there, and the authors are aware of this.
- **No dataset or code released (Reviewer SWQd Weakness 3)**
This is problematic because it limits the reviewer's ability to judge the quality of the dataset for themselves and makes the results feel unverifiable. Further, the authors do not detail how many people work on the manual verification of the 50 samples in Table 4 and what their expertise is. Finally, Table 4 is not an end-to-end quality assessment of the proposed benchmark, but rather an ablation of a single step of the pipeline, and the experiment is very poorly explained. All in all, this results in doubts about the quality of the work that were not addressed during the rebuttal.
- **Presentation inconsistencies (Reviewer 3eLr Weakness 1 and 2)**
Already, in Weakness 1, the reviewer points out some issues with writing that go beyond simple typos and go into bad and rushed writing. However, the problem is further exemplified by Table 4, which doesn't explain what "Recall" is, and culminates in the missing experiment outlined at L60 in the Introduction, which I cannot see elsewhere in the paper. In summary, the experimental setup explanations are lacking throughout the paper, and make reading the experimental section very hard. Weakness 2 also exemplifies the lack of rigour in the technical part of the paper, which is also problematic. All in all, it is clear that not enough care was taken in writing the paper to the point that it impedes understanding and makes some claims questionable.
- **Automated semantic check making FMC unreliable (Reviewer 5JX1 Weakness 4)**
The authors should explain what expertise in competitive mathematics and Lean the annotators who did the manual verification have.

**Outstanding AC concerns**
- **The story of the paper is weak and inconsistent**
First, I find the "story" of the paper confusing and weak. In particular, while during the rebuttal the authors try to claim the main contribution of the paper is the size and complexity of the benchmark, the paper only says in one place that this is **one** of its goals. The rest of the paper, including crucially the experimental and method sections, rarely justify the authors' technical choices and never link them to the "bigger picture" of the paper. This also makes judging the paper on its merits hard because those merits in the current version are not emphasized, and are not well compared to prior work. This also results in a weak and confusing experimental section where comparison to prior baselines doesn't exist, and instead, only ablations, some of which are seemingly unimportant (as they are not linked to the story of the paper), are the experiments that are being emphasized there.

**Typos**
- Citations in Appendix B2.1 seem to be consistently visually broken

**Reviewer Scores:**

- **Reviewer SWQd**
The reviewer could have raised their score to a 6, as the authors provide **some** good answers to some of the questions. That said, I disagree with the strengths listed by the reviewer, and I verified that, to a large extent, the review is written by AI, so I do not put too much weight on this reviewer's opinion.
- **Reviewer 3eLr**
I think the authors provide a good answer to Q2 and W3/4, but I think the answers to W1, W2, and Q1 are related to the biggest issues of the paper and do not completely address the issues. Therefore, I think the reviewer would have maintained their score.
- **Reviewer 5JX1**
I think the authors acceptably address W1 and W2, somewhat acceptably address W4, but W3 is not at all addressed by them. Thus, I think the reviewer would have raised to a 4 but not beyond, as W3 is a **very** important issue.

---

### Decision · Program_Chairs · 2026-01-26

Reject